# Genetically manipulating endogenous Kras levels and oncogenic mutations in vivo influences tissue patterning of murine tumorigenesis

Özgün Le Roux[1], Nicole LK Pershing[1†], Erin Kaltenbrun[1‡], Nicole J Newman[1§], Jeffrey I Everitt[2], Elisa Baldelli[3], Mariaelena Pierobon[3], Emanuel F Petricoin[3], Christopher M Counter[1]*

[1]Department of Pharmacology & Cancer Biology, Duke University Medical Center, Durham, United States; [2]Department of Pathology, Duke University Medical Center, Durham, United States; [3]Center for Applied Proteomics and Molecular Medicine, School of Systems Biology, George Mason University, Manassas, United States

**\*For correspondence:**
chris.counter@duke.edu

**Present address:** †Departments of Pathology and Pediatric Infectious Disease, University of Utah, Salt Lake City, United States; ‡Nighthawk Biosciences Inc, Morrisville, United States; §West Virginia School of Osteopathic Medicine, Lewisburg, United States

**Abstract** Despite multiple possible oncogenic mutations in the proto-oncogene *KRAS*, unique subsets of these mutations are detected in different cancer types. As *KRAS* mutations occur early, if not being the initiating event, these mutational biases are ostensibly a product of how normal cells respond to the encoded oncoprotein. Oncogenic mutations can impact not only the level of active oncoprotein, but also engagement with proteins. To attempt to separate these two effects, we generated four novel Cre-inducible (LSL) *Kras* alleles in mice with the biochemically distinct G12D or Q61R mutations and encoded by native (nat) rare or common (com) codons to produce low or high protein levels. While there were similarities, each allele also induced a distinct transcriptional response shortly after activation in vivo. At one end of the spectrum, activating the *Kras*^LSL-natG12D allele induced transcriptional hallmarks suggestive of an expansion of multipotent cells, while at the other end, activating the *Kras*^LSL-comQ61R allele led to hallmarks of hyperproliferation and onco-genic stress. Evidence suggests that these changes may be a product of signaling differences due to increased protein expression as well as the specific mutation. To determine the impact of these distinct responses on RAS mutational patterning in vivo, all four alleles were globally activated, revealing that hematolymphopoietic lesions were permissive to the level of active oncoprotein, squamous tumors were permissive to the G12D mutant, while carcinomas were permissive to both these features. We suggest that different KRAS mutations impart unique signaling properties that are preferentially capable of inducing tumor initiation in a distinct cell-specific manner.

## Editor's evaluation

This article addresses the long-standing question of why specific mutations in RAS are associated with tumors arising in distinct tissues; whether this reflects biological expression or functionality. The authors utilize a clever mouse genetic approach to modulate both which Kras mutation is expressed and how much of that Kras mutant is expressed, expressing the same Kras mutation at either a low level (due to rare codon utilization) or at a higher level (through common codon utilization). This study convincingly supports the conclusion that different KRAS mutations impart unique signaling properties that are preferentially capable of inducing tumor initiation in a distinct cell-specific manner.

## Introduction

Why specific driver mutations track with different cancers is unknown, yet speaks to the very origins of cancer, with implications for early detection and prevention. This is particularly well illustrated with the RAS family of small GTPases, comprised of KRAS, NRAS, and HRAS. Focusing on KRAS, the most commonly mutated of the three (*Prior et al., 2020*), single-point mutations at one of three hotpot positions (G12, G13, and Q61) result in six possible substitutions that inhibit the intrinsic or extrinsic GTPase activity of the protein. These mutations render KRAS constitutively GTP-bound and active, which is well known to be oncogenic (*Simanshu et al., 2017*). Although this amounts to 18 possible oncogenic mutations, specific subsets tend to be found in different cancer types. For example, the most common oncogenic KRAS mutation is G12C in non-small cell lung cancer but Q61H in plasma cell myeloma (*Prior et al., 2020*). Mice similarly exhibit a bias of specific oncogenic Kras mutations towards different cancer types as different mutant *Kras* alleles have different tumorigenic potential when activated in different tissues (*Li et al., 2018*; *Poulin et al., 2019*; *Winters et al., 2017*; *Wong et al., 2020*; *Zafra et al., 2020*). Different oncogenic mutations also affect the ability of Nras to induce tumorigenesis in mice, as do the same mutations in different RAS isoforms (*Burd et al., 2014*; *Haigis et al., 2008*; *Kong et al., 2016*; *Murphy et al., 2022*). While this tissue 'tropism' of cancers towards specific RAS mutations has been appreciated for decades (*Bos, 1989*), the underlying mechanism is unclear.

Variation in the ability of specific KRAS mutants to be tumorigenic (or not) in different tissues ostensibly results from differences in oncogenic signaling between mutants and the response of normal cells thereof. Generally speaking, oncogenic signaling is a product of the amplitude of the signal (quantitative signaling) and/or the pathways engaged (qualitative signaling). In terms of quantitative signaling, different mutations can exhibit different degrees of activation (GTP-loading) and/or different sensitivities to positive (Ras GTP exchange factors [RasGEFs]) or negative (RAS GTPase activating proteins [RasGAPs]) regulators (*Gebregiworgis et al., 2021*; *Munoz-Maldonado et al., 2019*; *Simanshu et al., 2017*). Various methods to manipulate quantitative Kras signaling, such as through modulating recombination rates (*Singh et al., 2021*), homozygous expression of the mutant allele (*Wang et al., 2011*), changing codon usage to increase protein expression (*Pershing et al., 2015*), additional pharmacological activation of the mitogen-activated protein kinase (MAPK) pathway (*Cicchini et al., 2017*), and so forth (*Li et al., 2018*), all affect tumorigenesis. On the other hand, perhaps one of the best examples of qualitative differences in KRAS signaling is the G12R mutant. Unlike the more canonical G12D mutation, the G12R mutant exhibits reduced the RAS effector PI3Kα binding and PI3K/AKT signaling (*Hobbs et al., 2020*). Activating an inducible *Kras*[LSL-G12R] allele in the pancreas led to fewer early premalignant lesions compared to the much more tumorigenic *Kras*[LSL-G12D] allele (*Zafra et al., 2020*). Despite an appreciation that different KRAS mutations can manifest in quantitative or qualitative signaling differences, how each contributes to the mutational patterns of this oncogene in cancer is unclear.

KRAS mutations are often initiating, being sufficient to induce tumorigenesis in mice and truncal in many human cancers. Thus, the bias of specific cancers towards distinct KRAS mutations could arise from potential tissue-specific differences of normal cells in the mutagenic process or repair capacity and/or in their responses to the nature of the signaling imparted by specific KRAS mutations (*Li et al., 2018*). In regard to the latter, determining the immediate response of normal cells to different KRAS mutations in vivo may help elucidate the mutational patterning of this oncogene. Identifying an oncogenic mutation arising in the *KRAS* gene from the cell-of-origin prior to becoming a tumor is challenging in humans. Mice, on the other hand, provide an ideal model system to experimentally explore this phenomenon as the point of tumor initiation can be precisely defined using inducible oncogenic *Kras* alleles. To therefore explore why different cancer types have a bias towards specific KRAS mutations, we created four novel inducible murine *Kras* alleles with different oncogenic mutations that were expressed at either low or high levels. In this way, we hypothesize that tissues in which tumor initiation is driven by the level of oncogenic signaling would preferentially develop tumors at higher protein levels while those driven by a specific mutant would preferentially develop tumors with just one of the two mutants.

We chose two completely different oncogenic mutations for these alleles, G12D and Q61R. G12D places a negatively-charged headgroup into the catalytic cleft of RAS and blocks extrinsic (RasGAP-mediated) GTPase activity (*Parker et al., 2018*). On the other hand, Q61R replaces the catalytic

amino acid with one that has a positively-charged headgroup, disrupting the position of the active site water molecule necessary for intrinsic GTP hydrolysis (*Buhrman et al., 2010*). Q61 is also essential for extrinsic GTP hydrolysis, as it stabilizes the transition state via hydrogen bonds to the γ-phosphate and nucleophilic water while providing another hydrogen bond to the RasGAP arginine finger (*Grigorenko et al., 2007*; *Kötting et al., 2008*; *Rabara et al., 2019*; *Scheffzek et al., 1997*). Comparing these two mutants directly reveals Q61R to have significantly lower GTP exchange and hydrolysis rates than G12D in Nras (*Burd et al., 2014*) and reduced RasGAP-mediated GTP hydrolysis rates in Kras (*Rabara et al., 2019*), akin to other substitutions at these two positions (*Gebregiworgis et al., 2021*; *Smith et al., 2013*). In those few cases in which the tumorigenic potential of the G12D and Q61R mutants of the same Ras isoform has been directly compared in mice, tissue-specific expression of the G12D mutant was less oncogenic than Q61R in both Nras-induced melanoma (*Burd et al., 2014*) and Kras-induced myeloproliferative neoplasm (*Kong et al., 2016*).

To alter the expression of these two mutants, the first three coding exons of *Kras* were fused and encoded by either their native rare codons, which is known to retard Kras protein expression, or common codons to increase Kras protein expression (*Ali et al., 2017*; *Fu et al., 2018*; *Lampson et al., 2013*; *Pershing et al., 2015*). We chose the novel approach of altering mammalian codon usage to modulate protein expression in vivo (*Li and Counter, 2021*; *Pershing et al., 2015*; *Sasine et al., 2018*) as no additional elements are required to change protein levels, providing a simple, reproducible, and uniform way of modulating *Kras* levels in mice and derived cell lines.

These four alleles were activated and immediately thereafter the in vivo transcriptome was determined in the lungs. This revealed that while these four alleles certainly shared similarities, each also induced unique transcriptional responses. Increased expression shifted the transcriptional hallmarks from those indicative of an expansion of multipotent cells to that of hyperproliferation and oncogenic stress. Changing the mutation type shifted the hallmark of estrogen response in the G12D mutant to that of the p53 pathway and DNA repair in the Q61R mutant. All four alleles were then globally activated, revealing that hematolymphopoietic lesions were permissive to the level of active oncoprotein, squamous tumors were preferentially permissive to the G12D mutant, while carcinomas tended to be permissive to both these changes. These findings support tissue-specific responses to the degree and type of signaling imparted by different oncogenic mutations molds the tissue tropism of cancers towards specific oncogenic KRAS mutations.

## Results

### A panel of inducible oncogenic *Kras* alleles designed to separate the effects of mutation type from the activity level of the oncoprotein

To explore how specific cancers are driven by distinct KRAS mutations, we reasoned that effects unique to a mutation could be separated from those imparted by activation level by simply comparing two different oncogenic mutants expressed at high or low levels. To this end, we created the four novel Cre-inducible oncogenic *Kras* alleles $Kras^{LSL-natG12D}$, $Kras^{LSL-natQ61R}$, $Kras^{LSL-comG12D}$, and $Kras^{LSL-comQ61R}$ (*Figure 1A*). Each began with an LSL transcriptional/translational repressor sequence (STOP) flanked by *lox*P sites (*Jackson et al., 2001*) engineered into the intron of *Kras* following the first non-coding exon to provide temporal and spatial control of gene expression. This was followed by a fusion of the first three coding exons encoded by either their native (*nat*) rare codons, which are known to retard protein expression, or 93 of these rare codons converted to their common (*com*) counterparts to increase protein expression (*Figure 1—figure supplement 1* and *Figure 1—source data 6*). Each version contained either a G12D or Q61R mutation. As noted above, these two mutants alter RAS activity in a biochemically different manner (*Munoz-Maldonado et al., 2019*; *Simanshu et al., 2017*), with Q61R reported to yield higher levels of active (GTP-bound) Ras (*Burd et al., 2014*; *Kong et al., 2016*; *Pershing et al., 2015*). This was followed by the next intron containing an FRT-NEO-FRT cassette for ES selection, which was excised via Flp-mediated recombination in the resultant animals, after which the *Flp* transgene was outbred. Finally, the remainder of the gene was left intact so as to generate the two Kras4a and Kras4b isoforms, as both contribute to tumorigenesis (*To et al., 2008*), potentially through unique protein interactions (*Amendola et al., 2019*).

To confirm that the expression and activity of these four versions of oncogenic *Kras* was reflected in their design, we ectopically expressed the corresponding FLAG-tagged murine *Kras* cDNAs in

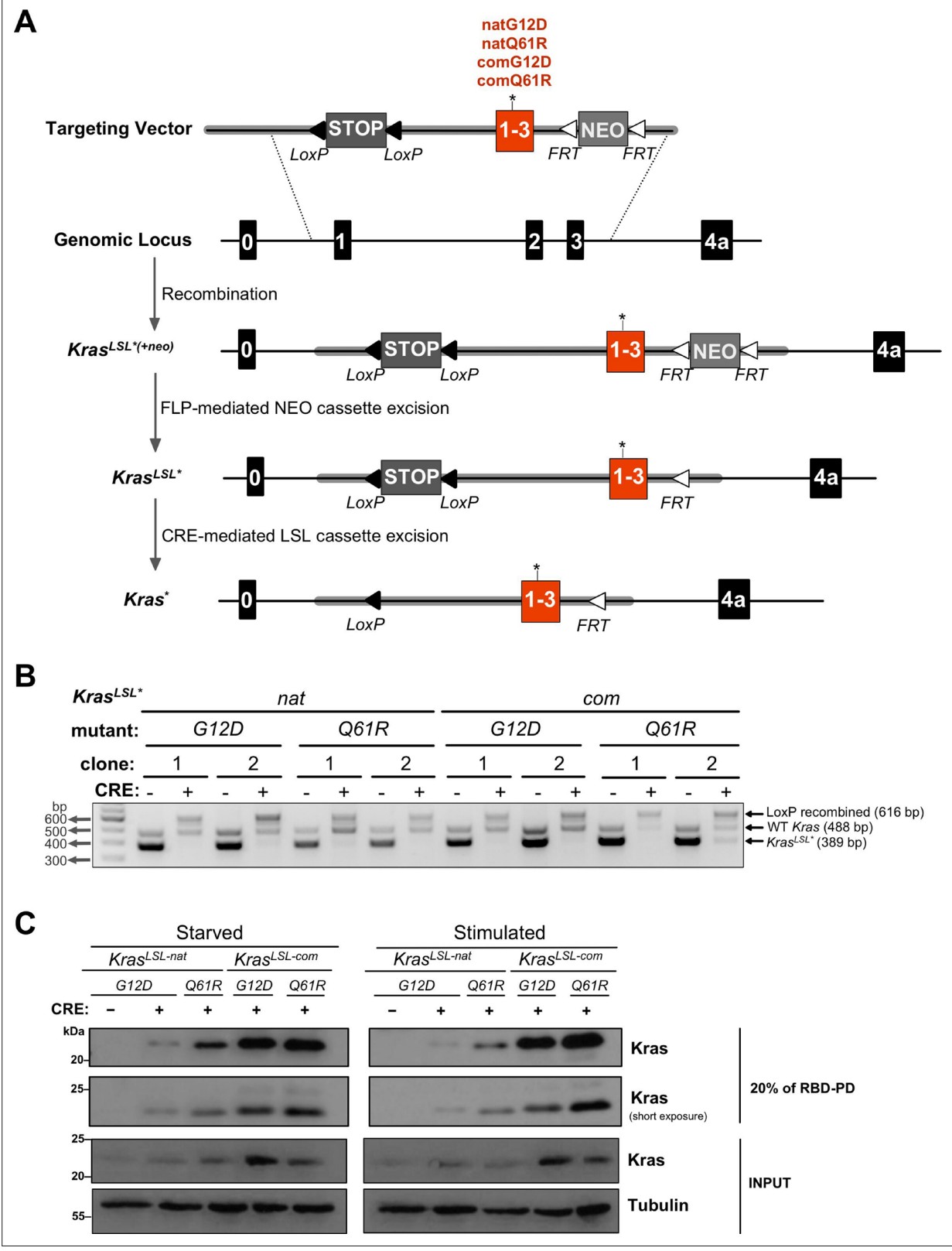

**Figure 1.** Conditional *Kras^LSL* alleles with different oncogenic mutations and codon usage. (**A**) Schematic of generating and activating *Kras^LSL* alleles with the first three coding exons fused and encoded by native (*nat*) versus common (*com*) codons with either a G12D or Q61R mutation. (**B**) PCR genotyping of two independently derived mouse embryonic fibroblast (MEF) cultures (two biological replicates) with the indicated *Kras* alleles in the absence and presence of Cre recombinase (CRE) to detect the unaltered wild-type *Kras* allele product (*WT*, 488 bp) and the unrecombined (*Kras^LSL\**,

*Figure 1 continued on next page*

**Figure 1 continued**

389 bp) and recombined (LoxP recombined, 616 bp) *Kras*^LSL^ allelic products. Gel images were cropped and color inverted for optimal visualization. Full-length gel images are provided in *Figure 1—source data 2*. (**C**) Expression levels, determined by immunoblot with an anti-Kras antibody, RAS activity levels, determined by RBD pull-down (RBD-PD) of lysates collected from MEF cells derived from mice with the indicated *Kras* alleles in the presence of Cre recombinase (CRE). MEF cultures derived from *Kras*^LSL-natG12D^/+ mice in the absence of Cre recombinase were used as negative control. MEF cultures were either serum starved overnight (starved) or serum starved overnight followed by serum stimulation for 5 min (stimulated). 20% of the elute from RBD-PD and 30 μg total protein from the total cell lysates were loaded. Tubulin serves as loading control. One of two biological replicates; see *Figure 1—figure supplement 3* for the second biological replicate. Full-length gel images are provided in *Figure 1—source data 3*.

The online version of this article includes the following source data and figure supplement(s) for figure 1:

**Source data 1.** Full-length gel images of RBD pull downs.

**Source data 2.** Full-length gel image of genotyping of mouse embryonic fibroblast (MEF) cultures derived from the *Kras*^LSL^ alleles in *Figure 1B*.

**Source data 3.** Full-length gel images of RBD pull-downs in MEFs.

**Source data 4.** Full-length gel images of Kras expression and activity using RBD pull-downs from lung tissue.

**Source data 5.** Ct values from the qRT-PCR analysis in *Figure 1—figure supplement 5*.

**Source data 6.** Sequence of coding exons 1 to 3 of the four *Kras*^LSL^ alleles.

**Figure supplement 1.** Codon usage of the four novel *Kras*^LSL^ alleles.

**Figure supplement 2.** Expression and activity level of proteins encoded by cDNA versions of the four *Kras*^LSL^ alleles.

**Figure supplement 3.** Second replicate of RBD pull-down assays from *Figure 1C*.

**Figure supplement 4.** Expression and activity levels in the lung upon activating each *Kras*^LSL^ allele.

**Figure supplement 5.** qRT-PCR validates an increase in RAS target gene expression with increased Kras activity.

**Figure supplement 6.** RPPA analysis of proteins and phosphoproteins within the MAPK signaling in the lung upon activation of *Kras*^LSL-natG12D/+^ versus *Kras*^LSL-comQ61R/+^.

HEK-HT cells that critically depend upon oncogenic RAS for tumorigenesis (*Hahn et al., 1999*). The total amount of ectopic Kras protein was then determined by immunoblot with an anti-FLAG antibody, and the level of GTP-bound and active Kras was determined by pull-down with a Ras binding domain (RBD) peptide followed by immunoblot or ELISA. Based on common codons increasing protein expression and Q61R leading to higher levels of GTP-bound RAS, the four constructs display the expected stepwise increase in RBD-bound Kras in the ascending order of Kras^natG12D^<Kras^natQ61R^ <Kras^comG12D^<Kras^comQ61R^ in both assays (*Figure 1—figure supplement 2*, *Figure 1—source data 1*).

## The effect of codon usage and mutation type on Kras activity in derived murine cells

To evaluate the endogenous functionality of this allelic set, we derived and characterized two separate mouse embryonic fibroblast (MEF) cultures from each of the genotypes *Kras*^LSL-natG12D/+^, *Kras*^LSL-natQ61R/+^, *Kras*^LSL-comG12D/+^, and *Kras*^LSL-comQ61R/+^, and validated that all four alleles could be recombined upon the expression of Cre recombinase (*Figure 1B*, *Figure 1—source data 2*). We then confirmed that this allelic set yields an increase in endogenous Kras activity consistent with codon usage and mutation type. These pairs of MEF cultures expressing Cre were either serum starved or stimulated to assess basal or stimulated Kras activity, respectively. RBD pull-down followed by immunoblot demonstrated the stepwise increase in Kras expression and activity levels in the ascending order of Kras^natG-12D^<Kras^natQ61R^ <Kras^comG12D^<Kras^comQ61R^ in both conditions (*Figure 1C*, *Figure 1—figure supplement 3*, *Figure 1—source data 3*). Thus, endogenous expression of these four alleles revealed that the Q61R mutant is more active than the G12D mutant, and changing rare codons to common increases the expression and activity of endogenous Kras oncoprotein.

## The effect of codon usage and mutation type on Kras activity in vivo

To validate these findings in vivo, we crossed the *Kras*^LSL-natG12D/+^, *Kras*^LSL-natQ61R/+^, *Kras*^LSL-comG12D/+^, and *Kras*^LSL-comQ61R/+^ genotypes into a *Rosa26*^CreERT2/+^ background, which expresses a tamoxifen-inducible Cre from the endogenous *Rosa26* promoter that is active in a broad spectrum of tissues (*Ventura et al., 2007*). Two adult mice from each of the four derived cohorts, as well as the control strain (*Rosa26*^CreERT2/+^), were injected with tamoxifen, and seven days later humanely euthanized, their lungs removed, and duplicate samples of derived protein lysates subjected to RBD pull-down followed by

immunoblot (*Figure 1—figure supplement 4A*). As in the case of the MEF cultures, we observed a stepwise increase in Kras expression and activity consistent with the codon usage and oncogenic mutation in this allelic set (*Figure 1—figure supplement 4B C*, *Figure 1—source data 4*). To assess whether this was reflected in the expression of downstream target genes, we generated the same four cohorts of mice, but in this case isolated RNA from the lungs of euthanized mice seven days after tamoxifen injection and determined the level of mRNA encoded by five known Ras target genes by qRT-PCR (*Figure 1—figure supplement 5A*). This revealed the expected increase in three such transcripts when *Kras* was encoded with common codons, which was further increased in the Q61R-mutant background (*Figure 1—figure supplement 5B C*, *Figure 1—source data 5*). To further validate this effect at the protein level, we performed reverse-phase protein array (RPPA) analysis of lung tissue upon activation of the two extreme alleles, $Kras^{LSL-natG12D}$ versus $Kras^{LSL-comQ61R}$ in the $Rosa26^{CreERT2/+}$ background, with $Rosa26^{CreERT2/+}$ serving as a control. Three mice from each genotypes were injected with tamoxifen and seven days later euthanized, their lungs removed, embedded, and derived protein lysates subjected to RPPA analysis. When normalized to the control tissue, there was a clear increase in the level of phosphorylated proteins of the MAPK pathway and downstream transcription factors upon activating the $Kras^{LSL-comQ61R}$ allele compared to activating the $Kras^{LSL-natG12D}$ allele (*Figure 1—figure supplement 6*, *Supplementary file 1*). Thus, the increase in Kras activity imparted by the Q61R mutant or by common codons observed in cultured cells is recapitulated in lung tissue.

## The effect of codon usage and mutation type on Kras biological activity in vivo

We next assessed whether these alleles were proportionally tumorigenic in a side-by-side comparison in the same organ. Each of these four alleles were crossed into a $CC10^{CreER/+}$ (and $Rosa26^{CAG-fGFP/+}$) background, and the resultant mice were injected with tamoxifen to specifically induce Cre-mediated recombination in the lung, after which every month thereafter for 6 months, five mice from each of the four cohorts were euthanized (*Figure 2—figure supplement 1A*). The lungs from all mice were visually analyzed for the presence of surface pulmonary tumors (*Figure 2—figure supplement 1B*), and in addition, two H&E-stained sections from pairs of mice were assayed for the presence and type of pulmonary tumors by a veterinarian pathologist blinded to the genotype (*Figure 2A*). This revealed a stepwise increase in early-onset and tumor burden of pulmonary lesions in lock step with the increased biochemical activity of the oncoproteins in the ascending order of Kras[natG12D]<Kras[natQ61R]<Kras[comG12D]<Kras[comQ61R] (*Figure 2A, B*). These tumorigenic phenotypes were the result of activating the inducible *Kras* alleles in the lungs as five control wild type ($Kras^{+/+}$) mice in the same background that were injected with tamoxifen failed to develop tumors after 13 months, twice the length of the study (*Figure 2—figure supplement 1C, D*). Finally, we find preliminary evidence in a few animals that the G12D-mutant alleles preferentially developed atypical alveolar hyperplasia (AAH), whereas bronchiolar hyperplasia/dysplasia (BH) lesions were more prevalent with the Q61R-mutant alleles (*Figure 2—figure supplement 1E F*). We conclude that this allelic set exhibits an increase in tumorigenic potential consistent with the activity of the encoded oncoproteins, but potentially may also display evidence of mutation-specific effects on tumorigenesis.

## The response of lung tissue to *Kras* alleles with different codon usage or mutations in vivo

As the different tumor types and grades that arose upon activation of each of these four inducible oncogenic *Kras* alleles are presumably a product of tumor initiation, understanding these effects presumably lies in how cells immediately respond to these different oncoproteins. We thus compared the immediate transcriptional response, as a measure of the cellular changes upon activation of each allele, in the lung. Cohorts of three adult mice from each of the four inducible oncogenic *Kras* alleles in $Rosa26^{CreERT2/+}$ background were injected with tamoxifen, and seven days later the animals were euthanized, their lungs removed, and RNA was isolated for bulk transcriptome sequencing (*Figure 3—figure supplement 1A*).

The resultant transcriptomes (GSE181628) exhibited a gradual increase in the number of genes differentially expressed with the level of active Kras (as assessed above by the RBD pull-down assay in lung tissue). Namely, we identified 7, 15, 73, and 701 genes were differentially expressed (with absolute $\log_2$ fold change larger than 2, and adjusted p-value less than 5%) upon activation of the

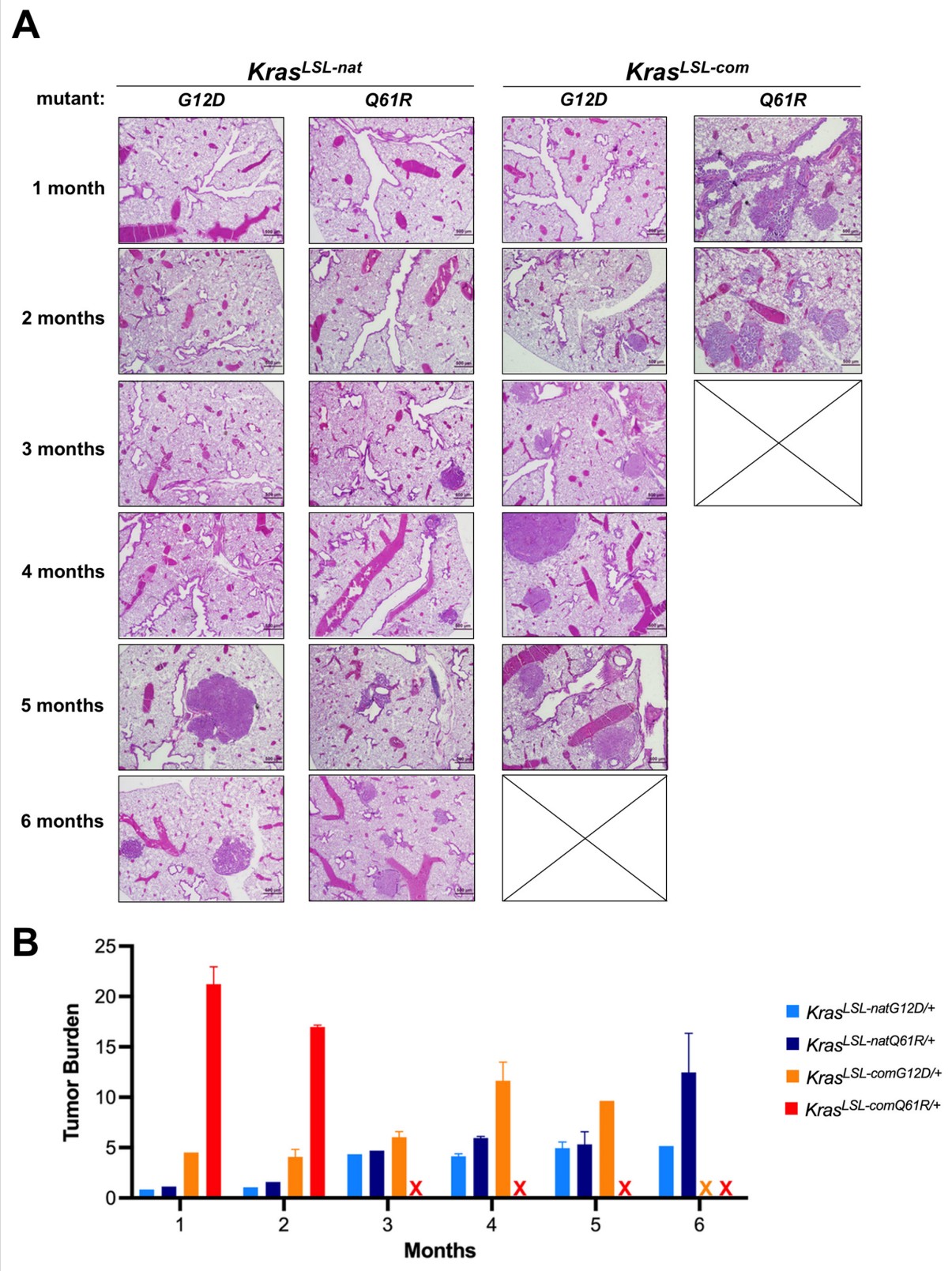

**Figure 2.** Biological effect upon activating each oncogenic *Kras* allele. (**A**) Examples of H&E-stained lung sections and (**B**) the mean ± SD % tumor burden from microscopic analysis of two lung sections from five mice with the indicated *Kras^LSL* alleles in a *CC10^CreER/+;Rosa26^CAG-fGFP/+* background at each of the indicated times post-tamoxifen injection.

*Figure 2 continued on next page*

*Figure 2 continued*

The online version of this article includes the following figure supplement(s) for figure 2:

**Figure supplement 1.** Tumor phenotypes track with intrinsic oncogenic Kras signaling.

*Kras*<sup>LSL-natG12D</sup>, *Kras*<sup>LSL-natQ61R</sup>, *Kras*<sup>LSL-comG12D</sup>, and *Kras*<sup>LSL-comQ61R</sup> alleles, respectively, compared to the wild-type *Kras* allele (*Figure 3A*, *Figure 3—figure supplements 2–6*, *Figure 3—source data 1*). Principal component analysis of these four transcriptomes revealed that the transcriptome upon activating both alleles encoded by native rare codons differed from both alleles encoded with common codons,while the G12D mutant was most distinct from the Q61R mutant in the *Kras*<sup>LSL-com</sup> background (*Figure 3—figure supplement 1B*). Similarly, when two mutants were compared, the ratio of uniquely differentially expressed genes specific to each allele increased from the G12D to the Q61R mutant when *Kras* was encoded with common codons (34–93%) more than it did when encoded with native rare codons (0–6%) (*Figure 3—figure supplement 6*, *Figure 3—source data 1*). Comparing the transcriptomes of the two *Kras* alleles encoded with native rare versus common codons revealed that the two native-encoded alleles decreased KRAS Signaling UP hallmarks while the two common-encoded alleles induced signaling events related to oncogenic stress, such as an increase in Oxidative Phosphorylation, Glycolysis, Reactive Oxygen Species, MTORC1 signaling, Peroxisome, Xenobiotic Metabolism, and UV Response UP hallmarks and a decrease in UV Response DN, Hedgehog Signaling, and Apical Junction hallmarks (*Figure 3B*, *Figure 3—figure supplement 1C*, *Figure 3—source data 2*). We interpret this as expression level largely dictating the degree of oncogenic signaling in this allelic set. Comparing the G12D versus Q61R transcriptomes revealed both G12D-mutant alleles induced the Estrogen Response Late hallmark while the Q61R-mutant alleles induced DNA Repair and P53 Pathway and decreased Interferon Alpha Response hallmarks, suggesting that there is a strong tumor-suppressive response uniquely activated by the Q61R-mutant alleles (*Figure 3C*, *Figure 3—figure supplement 1C*, *Figure 3—source data 2*).

To validate the transcriptional responses detected by bulk RNA-seq analysis, we quantified four to six marker genes in a subset of selected hallmarks. Two adult mice from each of the four cohorts and control mice were injected with tamoxifen as above, and seven days later the animals were euthanized, their lungs removed, RNA isolated, and the level of select transcripts determined by qRT-PCR. We identified similar expression patterns to the transcriptome signature in the hallmarks of TNFα Signaling via NFκB and Interferon-γ, which increase with Kras activity (*Figure 3—figure supplement 7A B*, *Figure 3—source data 3*), and EMT and Myogenesis, as these hallmarks were enriched in G12D mutants but depleted in Q61R mutants (*Figure 3—figure supplement 7A C*, *Figure 3—source data 3*).

To probe these transcriptomes for evidence of an orchestrated response of normal cells to different Kras mutants, we repeated transcriptome analysis of lung tissue on the two extreme cases, namely, *Kras*<sup>LSL-natG12D</sup> versus *Kras*<sup>LSL-comQ61R</sup> (GSE181627). Activation of the *Kras*<sup>LSL-natG12D</sup> allele resulted in transcriptional signatures suggestive of an expansion of multipotent cells (*Figure 3—figure supplement 8A B*). Namely, Gene Set Enrichment Analysis (GSEA) hallmarks indicative of Epithelial Mesenchymal Transition (EMT, TGFβ Signaling, Wnt/βcatenin Signaling, Notch Signaling, Apical Junction, Apical Surface, and Estrogen Response Early) and multiple cell lineages (Adipogenesis, Myogenesis, Hedgehog Signaling, and Pancreas β Cells). Conversely, activation of the *Kras*<sup>LSL-comQ61R</sup> allele had all features of a potent oncogenic signaling leading to hyperproliferation and oncogenic-induced stress. Namely, GSEA hallmarks indicative of high oncogenic signaling (KRAS Signaling UP) and unrestrained proliferation (MYC Targets V1, and E2F Targets), leading to reactive oxygen species (Oxidative Phosphorylation, and Reactive Oxygen Species Pathway) and a DNA damage response (DNA Repair, P53 Pathway, and G2M Checkpoint) followed by apoptosis (Apoptosis) and inflammation (Inflammatory Response, IL6 JAK STAT3 Signaling, TNFα Signaling via NF-κB, and Allograft Rejection).

To assess the response of cells at the protein level, we performed RPPA analysis of lung tissue from these same two genotypes. This revealed that the level of protein/phosphoproteins of RAS/MAPK, PI3K/AKT, Growth, DNA repair, senescence/autophagy/apoptosis, and IL/Jak/Stat pathways was higher upon activation of the *Kras*<sup>LSL-comQ61R</sup> compared to the *Kras*<sup>LSL-natG12D</sup> allele (*Figure 3—figure supplement 9* and *Supplementary file 1*), consistent with the transcriptome analysis. Finally, STRING analysis of the top ten genes from each of these GSEA hallmarks revealed potential crosstalk between the various signaling pathways, suggesting a consolidated signaling program by the two different

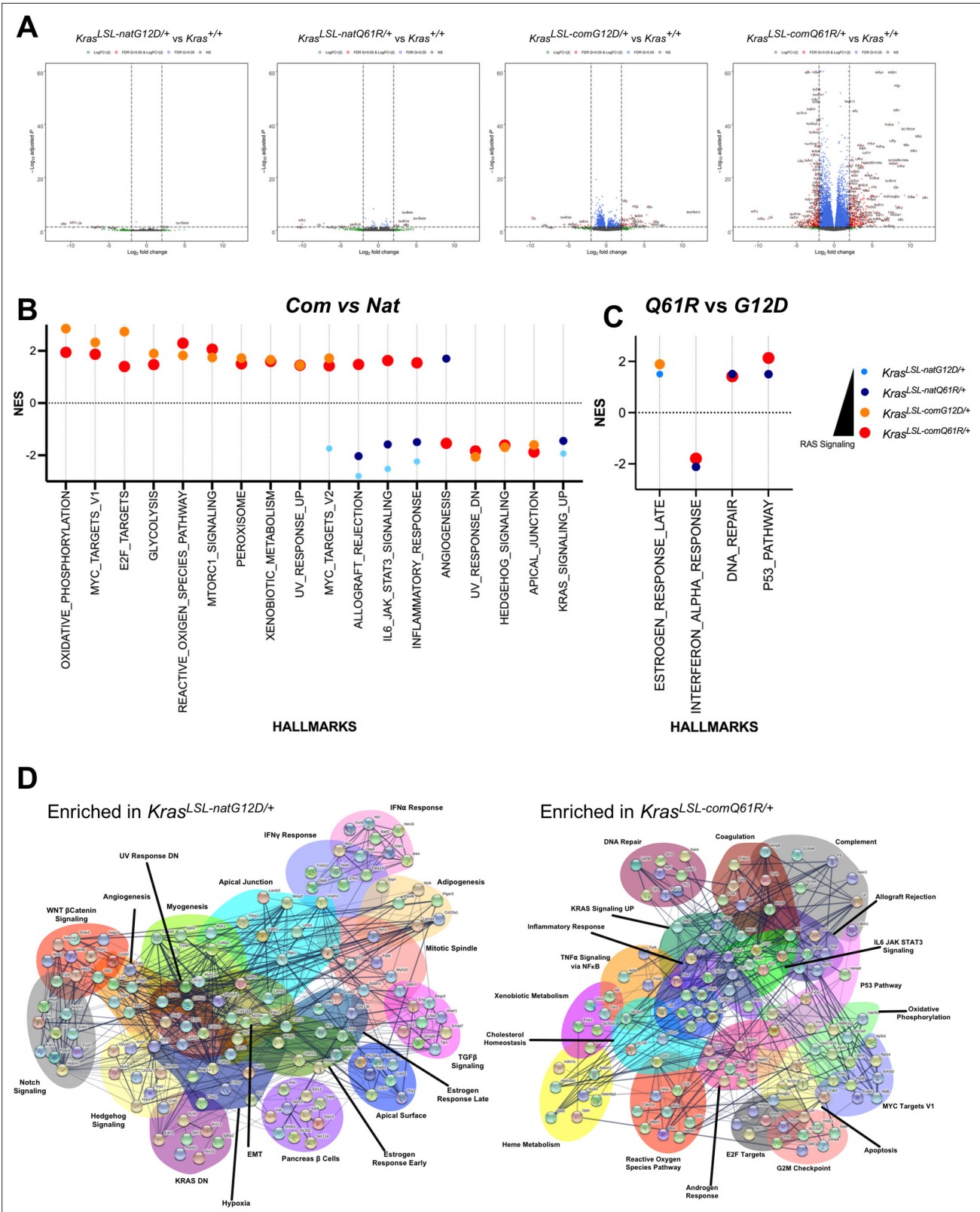

**Figure 3.** Lung transcriptome induced by each oncogenic *Kras* allele. (**A**) Volcano plot of the log₂ fold change versus p-value of the genes showing differential expression in each allele compared to the wild-type (+/+) *Kras* allele in the lung. Full-sized plots are provided in *Figure 3—figure supplements 2–5*. (**B, C**) Normalized enrichment score and patterns of the indicated Gene Set Enrichment Analysis (GSEA) hallmarks differentially enriched by *Kras* codon usage (expression) (**B**) and mutation type (**C**). Only hallmarks with a false discovery rate (FDR) < 5% are shown. Dot size is

*Figure 3 continued on next page*

*Figure 3 continued*

adjusted to RAS activity for better visualization. All GSEA hallmarks differentially enriched upon activating *Kras^LSL* alleles with an FDR < 5% are provided in *Figure 3—figure supplement 7*. (**D**) STRING analysis of the top ten genes in the differentially enriched GSEA hallmarks identified in RNA-seq analysis of the lungs of *Rosa26^CreERT2/+*;*Kras^LSL-natG12D/+* versus *Rosa26^CreERT2/+*;*Kras^LSL-comQ61R/+* mice seven days after tamoxifen injection.

The online version of this article includes the following source data and figure supplement(s) for figure 3:

**Source data 1.** Differentially expressed genes in each allele from *Figure 3—figure supplements 2–6*.

**Source data 2.** Normalized enrichment scores of the hallmarks identified in *Figure 3—figure supplement 1C*.

**Source data 3.** Ct values from the qRT-PCR analysis in *Figure 3—figure supplement 7*.

**Figure supplement 1.** Lung transcriptome upon activating each *Kras^LSL* allele.

**Figure supplement 2.** Lung transcriptome upon activating the *Kras^LSL-natG12D* allele.

**Figure supplement 3.** Lung transcriptome upon activating the *Kras^LSL-natQ61R* allele.

**Figure supplement 4.** Lung transcriptome upon activating the *Kras^LSL-comG12D* allele.

**Figure supplement 5.** Lung transcriptome upon activating the *Kras^LSL-comQ61R* allele.

**Figure supplement 6.** Venn diagram of the number of shared differentially expressed genes in the lung transcriptome upon activation of the *Kras^LSL* alleles.

**Figure supplement 7.** Validation of signaling responses via qRT-PCR analysis of the selected genes.

**Figure supplement 8.** Comparison of the transcriptomes upon activating the *Kras^LSL-natG12D* versus *Kras^LSL-comQ61R* allele.

**Figure supplement 9.** RPPA analysis of RAS downstream pathways upon activating the *Kras^LSL-natG12D* versus *Kras^LSL-comQ61R* allele.

oncoproteins manifests in very different responses by normal cells (*Figure 3D*). We suggest that this allelic set moves the response of normal lung cells from an expansion of multipotent cells to one of extreme oncogenic signaling and stress, with the type of oncogenic mutation potentially further modulating these responses.

## Tissue sensitivities to the different oncogenic *Kras* alleles

We next addressed the 'tropism' of tissues towards specific Kras mutants by determining the tumor landscape upon globally activating each allele. As each allele is expected to have a different oncogenic potential, we opted for a moribundity endpoint, as opposed to a fixed endpoint, to identify the tissues most permissive to tumorigenic conversion in a competition-based approach. To this end, each allele was again activated by tamoxifen using the aforementioned ubiquitous Cre driver *Rosa-26^CreERT2*. Mice were regularly monitored for moribundity endpoints, indicative of ensuing mortality due to cancer, at which time the animals were euthanized (*Figure 4—figure supplement 1*). As a control to rule out a tumor phenotype being a product of variations in gene activation, we validated Cre-mediated recombination between all four alleles across 14 diverse tissues seven days after tamoxifen injection. Only the ovary displayed reduced recombination, and hence was not included in the study (*Figure 4—figure supplement 2*, *Figure 4—source data 1*, and *Supplementary file 2*).

Plotting the percent survival for each of the four genotypes by the Kaplan–Meier approach revealed that the median life span of mice progressively increased from 14 days upon activating the *Kras^LSL-comQ61R* allele to 150 days upon activating the *Kras^LSL-natG12D* allele (*Figure 4A*). Not surprisingly, the number of tumors per animal mirrored these survival differences (*Figure 4—figure supplement 3*). Pairwise comparisons revealed that median survivals were statistically different, except between the *Kras^LSL-natG12D/+* versus *Kras^LSL-natQ61R/+* cohorts (*Figure 4—figure supplement 4* and *Supplementary file 3*). Of note, median survival was significantly decreased in mice with when *Kras* was encoded with common compared to native rare codons. Survival was also decreased in the mice with the Q61R compared to the G12D mutation when the allele contained the same codon usage (*Kras^LSL-natQ61R* versus *Kras^LSL-natG12D* and *Kras^LSL-comQ61R* versus *Kras^LSL-comG12D*). As both common codons and the Q61R mutation increase Kras activity, we suggest that the level of active oncoprotein is the dominant determinant of cancer survival in this model.

To identify the tissues permissive to tumorigenesis by each of these different alleles, eight different organs were removed from the mice during necropsy and analyzed for pathological changes as above. While there was much overlap, the prevalence and severity of specific cancer types varied between alleles, arguing that the different alleles can lead to differences in the tumor landscape (*Figure 4B*, *Figure 4—figure supplement 5*). This manifested in four general patterns of tissue permissivity:

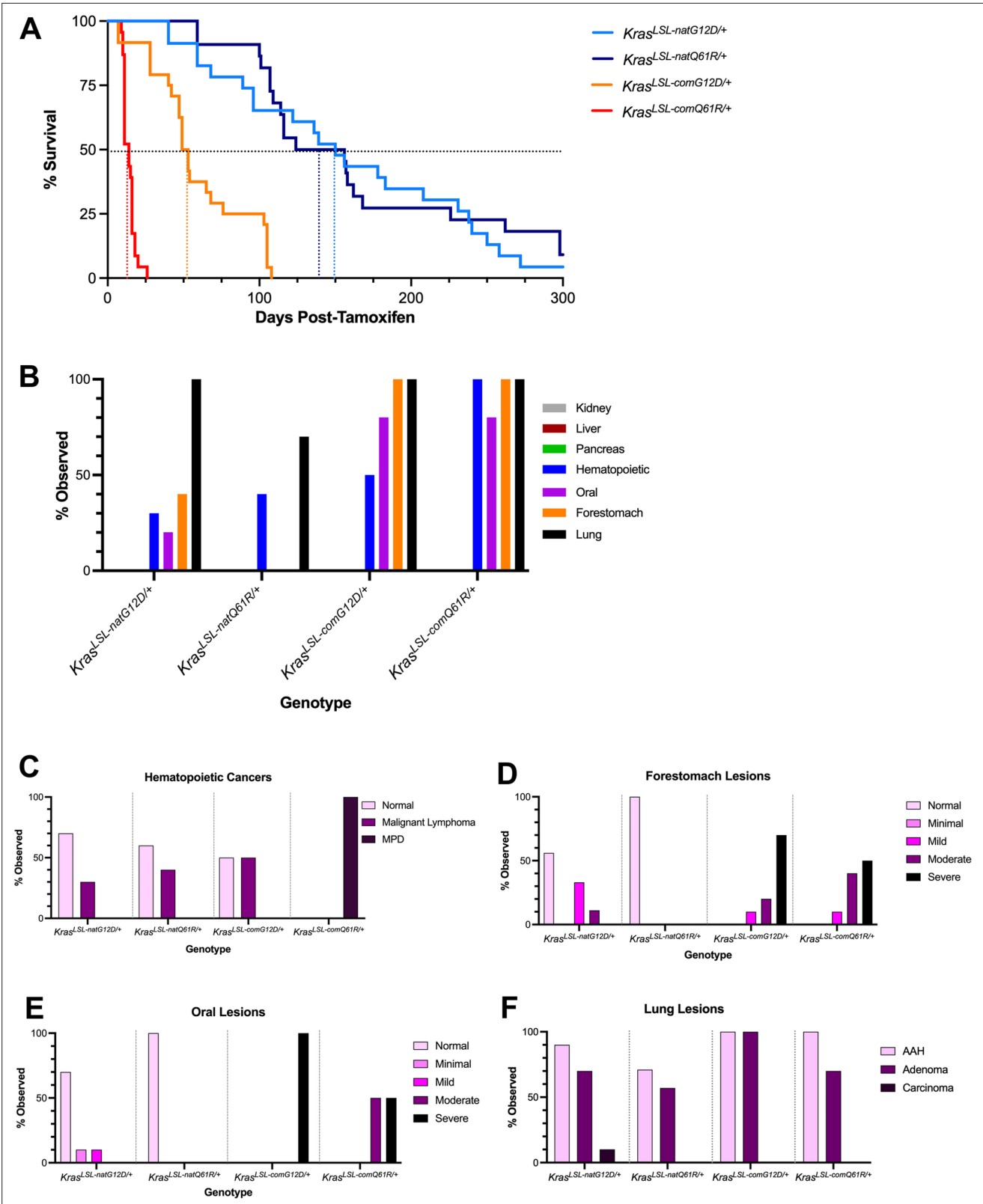

**Figure 4.** Tissue atlas of sensitivities to each oncogenic *Kras* allele. (**A**) Kaplan–Meier survival curve of the mice with the indicated *Kras^LSL* alleles after activation by tamoxifen. Dotted lines: 50% survival. (**B**) Number of mice with the indicated number of different tumor types. Examples of H&E-stained slides of the indicated tissues are provided in **Figure 4—figure supplement 5**. (**C–F**) Percentage of the indicated grades of hematolymphopoietic (**C**),

*Figure 4 continued on next page*

*Figure 4 continued*

forestomach (**D**), oral (**E**), and lung (**F**) lesions at moribundity endpoint in *Rosa26*$^{CreERT2}$/+ mice (n = 8–10) with one of the four indicated *Kras*$^{LSL}$ alleles after activation by tamoxifen.

The online version of this article includes the following source data and figure supplement(s) for figure 4:

**Source data 1.** Full-length gel images of the recombination PCRs, in *Figure 4—figure supplement 2B*.

**Figure supplement 1.** Global activation of each engineered *Kras*$^{LSL}$ allele using a ubiquitous Cre driver.

**Figure supplement 2.** Tissue-specific recombination rates of each oncogenic *Kras*$^{LSL}$ allele upon global activation.

**Figure supplement 3.** Number of tumor types induced upon globally activating each oncogenic *Kras*$^{LSL}$ allele.

**Figure supplement 4.** Pairwise comparisons of survival upon globally activating each oncogenic *Kras*$^{LSL}$ allele.

**Figure supplement 5.** Proliferative lesions induced upon activating each oncogenic *Kras*$^{LSL}$ allele.

**Figure supplement 6.** Histopathological analysis of pulmonary lesions induced upon activating each of the oncogenic *Kras*$^{LSL}$ alleles.

**Figure supplement 7.** The tumor landscape of *Rosa26*$^{CreERT2/+}$;*Kras*$^{LSL-rareG12D/+}$ mice after tamoxifen injection.

hematolymphopoietic neoplasias increased in severity with the level of active oncoprotein, squamous tumors were preferentially induced by the G12D-mutant alleles, carcinomas were induced by all four alleles, while many organs were resistant to oncogenic RAS-driven tumorigenesis within the time frame of the study (*Figure 4B*, *Figure 4—figure supplement 5*, and *Supplementary file 4*).

## A tissue sensitive to Kras activity

The incidence of hematolymphopoietic neoplasias increased with the biochemical activity of the Kras oncoprotein, as determined from analysis of MEF cultures and lung tissue. In more detail, the spleen and thymus from eight to ten mice from each of the four cohorts were removed at necropsy, after which H&E-stained sections were assayed for the presence and grade of hematolymphopoietic neoplasias as above. Beginning with the least active oncoprotein, most of the *Kras*$^{LSL-natG12D}$ mice after tamoxifen injection had no evidence of hematolymphopoietic neoplasms, although some mice had pathological features consistent with malignant lymphoma. The incidence of malignant lymphoma increased with *Kras*$^{LSL-natQ61R}$ allele, and then again upon activating the *Kras*$^{LSL-comG12D}$ allele. Pathological analysis also revealed medullary hyperplasia in the thymus, as well as leukemic infiltrates in the kidneys and pancreas of these latter mice, suggesting further progression of the lymphomas with this allele. Lastly, activating the *Kras*$^{LSL-comQ61R}$ allele induced severe myeloproliferative disease (MPD) with 100% penetrance, with extensive myeloproliferative infiltrates throughout many tissues (*Figure 4B, C*, *Figure 4—figure supplement 5*, *Supplementary file 4 and 5*, and not shown). While we cannot discount high oncogenic activity leading to a different hematopoietic disease in these later mice, we suggest that the incredibly short latency of the onset of severe systemic myeloid neoplasia may instead preclude development of longer latency tumors, such as lymphopoietic neoplasms. Thus, with this proviso, we suggest that hematolymphopoietic neoplasia are sensitive to the level of oncogenic activity, being induced at the lowest level of active Kras and progressively becoming more aggressive with increased activity.

## Tissues sensitive to Kras mutation type

Proliferative lesions of forestomach and oral squamous epithelium were preferentially induced by the oncogenic G12D mutant of Kras encoded by either native or common codons. In the case of the forestomach tumors, pathological analysis performed as above revealed that activation of the *Kras*$^{LSL-natG12D}$ allele induced squamous hyperplasia as well as mild and moderate grades of 'atypical' or dysplastic squamous lesions in the forestomach mucosa. Conversely, we did not detect any squamous proliferative changes upon activating the *Kras*$^{LSL-natQ61R}$ allele (*Figure 4B and D*, *Figure 4—figure supplement 5*, and *Supplementary files 4 and 5*), despite the previously detected higher activity of the Kras$^{natQ61R}$ oncoprotein in MEFs and lung tissue. The same trend was observed when *Kras* was encoded with common codons, except the difference was less extreme between the two mutants, and shifted towards more aggressive disease. Specifically, activating the *Kras*$^{LSL-comG12D}$ allele induced more severe grades of forestomach squamous lesions while activation of the *Kras*$^{LSL-comQ61R}$ allele now induced lesions that were of moderate grades (*Figure 4B and D*, *Figure 4—figure supplement 5*, and *Supplementary file 5*). Similar analysis of oral tumors revealed that activating the *Kras*$^{LSL-natG12D}$

allele induced minimal to mild grade squamous lesions, while activating the *Kras*<sup>LSL-natQ61R</sup> allele was not tumorigenic. Again, when Kras was encoded with common codons there was a shift to a more aggressive disease. Namely, activating the *Kras*<sup>LSL-comG12D</sup> allele induced severe squamous papilloma in all mice, while activating the *Kras*<sup>LSL-comQ61R</sup> allele induced a mixture of moderate and severe grade squamous papillomas (*Figure 4B and E*, *Figure 4—figure supplement 5*, and *Supplementary file 5*). As such, in these two organs, the G12D mutant is associated with more severe phenotypes than the Q61R mutant, and changing codon usage to increase expression shifts this difference to a more advanced stage.

## Tissues sensitive to both Kras oncogenic activity and mutation type

Pathological analysis performed as above in the lung revealed that the G12D mutation consistently induced more AAH and/or adenomas than the Q61R mutation when Kras was encoded with either common or native rare codons, both in terms of the number of animals with these lesions and the total number of these lesions per animal (*Figure 4B and F*, *Figure 4—figure supplements 5; 6A*, and *Supplementary file 5*). As was the case with forestomach and oral lesions, converting rare codons to common amplified the severity of lesions detected, which was particularly evident in the confluence of large peripheral AAH lesions induced preferentially by the*Kras*<sup>LSL-comG12D</sup> allele (*Figure 4B and F*, *Figure 4—figure supplements 5; 6B*). However, no BH lesions were induced by either mutant when *Kras* was encoded with native rare codons and instead were only prevalent when *Kras* was encoded with common codons. Further, the number of animals with BH lesions was higher upon activating the *Kras*<sup>LSL-comQ61R</sup> compared to the *Kras*<sup>LSL-comG12D</sup> allele (*Figure 4—figure supplements 5; 6C*). Assuming that AAH and BH lesions represent different types of tumors (and not different stages of the same tumor type), we suggest that tumorigenesis in the lung is influenced by both the degree of Kras activation and the mutation type, although temporal analysis argues (*Figure 2*) that the level of Kras activation nevertheless dominates tumorigenesis in this tissue.

## Tissues resistant to oncogenic Kras

Despite widespread tumorigenesis, we note that no overt proliferative lesions were detected at necropsy or by histopathological analysis in the pancreas, kidney, or liver (*Figure 4—figure supplement 5* and *Supplementary file 4*). A gross survey of other organs such as the colon, intestine, heart, skin, and mammary glands similarly failed to reveal macroscopically detectable tumors (not shown). In agreement, many of these same tissues, including pancreatic, were reported to be refractory to tumorigenesis upon activating a *Kras*<sup>LSL-G12D</sup> allele in the adult mice by CreER expressed from the *Rosa26* (*Parikh et al., 2012*; *van der Weyden et al., 2011*), CK19 (*Ray et al., 2011*), or *Ubc9* (*Matkar et al., 2011*) loci. Thus, many organs appear to be intrinsically resistant to the tumorigenic effects of oncogenic Kras, regardless of the mutation type or expression levels tested, at least within the time frame of this study and in the tested *Rosa26*<sup>CreERT2/+</sup> background.

# Discussion

Here, we describe the effect of activating four inducible oncogenic *Kras* alleles encoding two very different mutants in a native rare versus common codon background to explore the mechanisms underlying the bias of specific oncogenic KRAS mutations towards distinct cancer types. We suggest that the unique tumor patterns arising from each of these mutant alleles implies that tissues differ in their sensitivities to quantitative and/or qualitative RAS signaling, both as a product of oncoprotein signaling and the cellular response thereof. We acknowledge four caveats to this approach. First, these four alleles were generated by fusing the first three coding exons, an artificial gene architecture, and hence can only be compared to themselves and not to other types of *Kras* alleles. Second, these alleles were induced by an injection of tamoxifen to activate CreER expressed from the *Rosa26* locus. Admittedly, this is an unnatural situation whereby oncogenic Kras is expressed all at once in a multitude of tissues, potentially perturbing homeostasis in the whole animal. Nevertheless, as the identical design was applied to all four alleles, comparisons can be made within this allelic set. Third, Kras activity of this allelic set was defined in MEFs and lung tissue. We therefore acknowledge that cell-type differences in regulatory feedback pathways (*Lake et al., 2016*; *Liu et al., 2018*) or codon-dependent expression (*Peterson et al., 2020*) could result in different levels of Kras expression, activation, or

signaling compared to that observed in MEF cultures and lung tissue, which were used to define the activity of each Kras oncoprotein. Fourth, by the nature of the experimental design, tissue types with slower tumor progression are underrepresented, and hence may respond differently to activation of the different *Kras* alleles when assessed more specifically.

With these limitations in mind, transcriptome analysis shortly after activating these alleles revealed similarities between the four, but also distinct transcriptional changes. This began with activation of the *Kras*[LSL-natG12D] allele, which induced EMT and differentiation hallmarks. EMT is known to promote stem cell-like fates (*Floor et al., 2011*; *Mani et al., 2008*) and activation of oncogenic Kras in the murine lung generates tumors with many tissue lineages (*Tata et al., 2018*). Taken together, we suggest that the transcriptional signature induced by the Kras[natG12D] oncoprotein reflects either reprogramming towards or an expansion of cells with multipotent characteristics. In support, single-cell transcriptome profiling of lung tumors induced by targeted delivery of AAV-Cre in an *Kras*[LSL-G12D/+];*Trp53*[fl/fl] background identified a distinct population with a mixed cellular identity (*Marjanovic et al., 2020*). At the other end of the spectrum, activating the *Kras*[LSL-comQ61R] allele induced transcriptional hallmarks consistent with overt oncogenic signaling. Thus, as the level of Kras biochemical activity increased, so did the transcriptional signatures of RAS signaling, which at its crescendo resulted in transcriptional signatures indicative of hyperproliferation and oncogenic stress. Nevertheless, within this overarching pattern of increased signaling we also find evidence for both mutation- and codon (expression)-specific transcriptional responses.

Globally activating these four alleles revealed four tumorigenic patterns. First, hematolymphopoietic neoplasias appeared to be largely driven by the amount of active Kras, as defined by RBD pull-down in both MEFs and lung tissue, as well as qRT-PCR of *Kras* target genes, transcriptome analysis, and RPPA analysis in lung tissue. Namely, these neoplasias were induced at the lowest level of oncogenic activity and increased in aggressiveness with increased Kras activity. To independently validate this result, we found that globally activating a 'super-rare' version of *Kras*[LSL-rareG12D] allele encoded by the rarest codons induced hematolymphopoietic neoplasias, although in a much-protracted time frame (*Figure 4—figure supplement 7* and *Supplementary file 4*). This further supports hematolymphopoietic tissues as being more permissive to oncogenic Ras activity, suggestive of a dependency on quantitative signaling. Second, proliferative lesions of forestomach and oral squamous epithelium were preferentially induced by G12D-mutant alleles. Such a finding points towards qualitative signaling differences potentially driving these tumors, with higher expression shifting the effects to more aggressive grades. Equally plausible, however, perhaps either a RasGEF or RasGAP is uniquely expressed in these tissues that preferentially interacts with or affects signaling of only one of these mutants, implying quantitative signaling unique to the G12D mutant in these specific tissues. In support, GEF-mediated GTP exchange is more rapid with a Q61L versus G12V mutant of Kras (*Smith et al., 2013*), the RasGEF SOS1 is inactive toward KRAS[G12R] (*Hobbs et al., 2020*), and KRAS[Q61H] has diminished sensitivity to SHP2 inhibitors when compared to G12 or G13 mutants (*Gebregiworgis et al., 2021*). Third, the lung was sensitive to both the activation level and mutation type, with the G12D mutants favoring AAH and adenomas while high Ras activity favored BH lesions, perhaps reflecting a different cell-of-origin for these different lesions (*Sutherland et al., 2014*; *Xu et al., 2014*). AAH lesions may also speak to mutation-specific signaling, as here is a case in which we document Q61R mutants are more active, yet G12D mutants are more tumorigenic. With the caveat that Kras-GTP levels were determined in the entire lung, which may not reflect the actual levels in the cell-of-origin for AAH lesion, this implies qualitative differences may underlie these specific lesions. Fourth, many organs failed to develop lesions. We acknowledge that the short life span of some of the tested mice may very well prevent longer latency tumors from developing, and even though *Rosa26*-restricted Cre expression activated the four inducible oncogenic *Kras* alleles in tissues that did not form tumors, Cre may not be expressed in the tumor cell-of-origin within these tissues. With these caveats mind, these data suggest that many tissue are intrinsically resistant to the tumorigenic effects of oncogenic Kras. Instead, the hematopoietic system, lungs, forestomach, and oral mucosa are unique in being permissive to the tumorigenic potential of Kras oncoproteins, again, however, within the confines of the experimental design. Interestingly, these tissue sensitivities share some similarity to that of humans, namely, both species have RAS-associated cancers in the lung, mouth, and hematopoietic system but

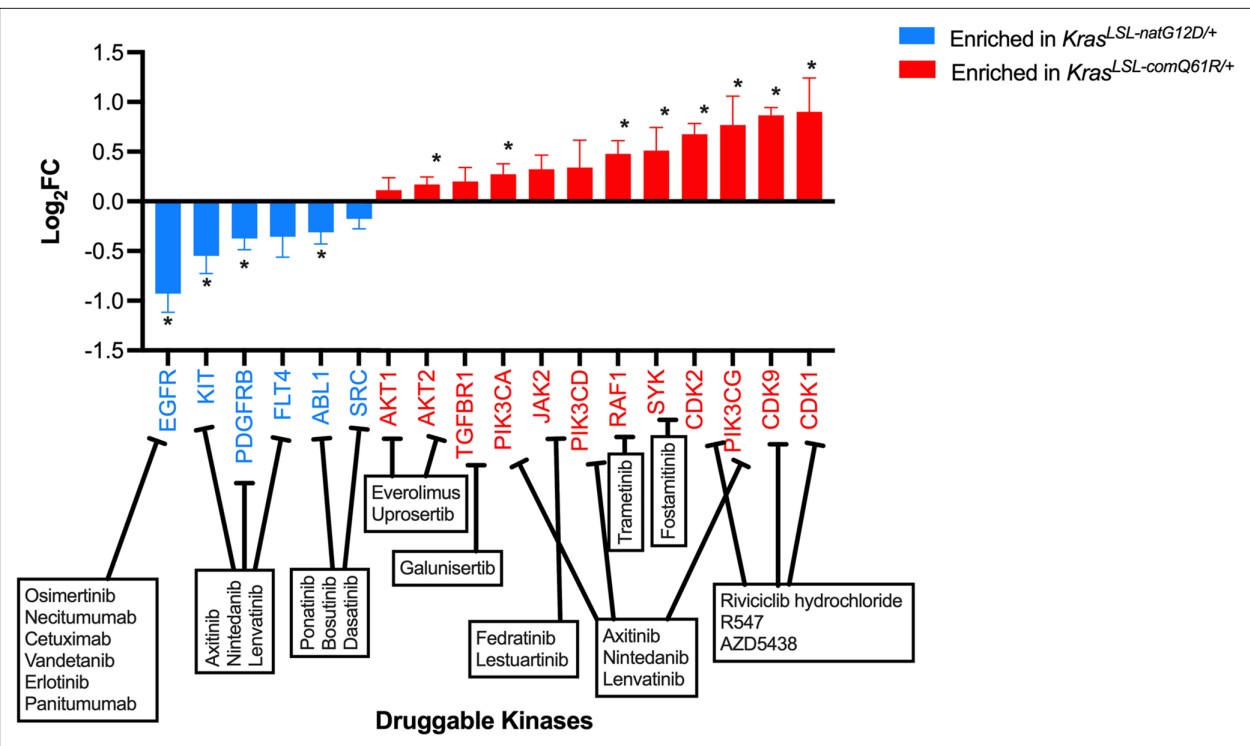

**Figure 5.** Transcriptome analysis predicts unique pharmacological vulnerabilities. Druggable kinases positively enriched in GSEA hallmarks identified in *Figure 3D* (blue *Kras*[LSL-natG12D/+], red *Kras*[LSL-comQ61R/+]). * Adjusted p-value<5%.

The online version of this article includes the following figure supplement(s) for figure 5:

**Figure supplement 1.** Differential expression of known transcription factors of genes enriched in the GSEA hallmarks identified in the transcriptome of lung tissue upon activating the *Kras*[LSL-natG12D] and *Kras*[LSL-comQ61R] alleles.

not in the mammary gland, skin, central nervous system, and so forth, but there is also some discordance (*Prior et al., 2020*).

How these different alleles drive the observed tumor patterns remains to be determined. However, comparing the two extreme cases – activation of the *Kras*[LSL-natG12D] versus the *Kras*[LSL-comQ61R] alleles in the lung – reveals transcription factors linked to the unique transcriptional responses of these two alleles. Specifically, we cross-referenced the transcriptome with a curated eukaryotic transcription factor database (*Matys et al., 2003*; *Matys et al., 2006*), identifying a set of upregulated transcription factors that tracked with the GSEA hallmarks of each oncoprotein (*Figure 5—figure supplement 1A, B* and *Supplementary file 6*). Namely, transcription factors tracking with GSEA hallmarks EMT, Estrogen Response Early, KRAS Signaling DN, and Apical Junction in the case of activating the *Kras*[LSL-natG12D] allele in the lung, and GSEA hallmarks TNFα Signaling via NFκβ, Complement, P53 Pathway, KRAS Signaling UP, and Inflammatory Response in the case of activating the *Kras*[LSL-comQ61R] allele, again in the lung (*Figure 5—figure supplement 1A, B*). Further, our finding that oncogenic RAS mutations are not identical in terms of activity (GTP-loading), oncogenic potential (induction of lung tumors), and cellular response (transcriptome and RPPA analysis) suggests the intriguing possibility that different initiating RAS mutations may have different therapeutic sensitivities. Indeed, again censoring the GSEA hallmarks in these two extreme cases for pharmacological targets already drugged in the clinic identified FLT4/PDGFRB/KIT, EGFR, and ABL1/SRC as specific to an activated *Kras*[LSL-natG12D] allele, while AKT1/2, PI3KCA/CD/CG, SYK, RAF1, JAK2, TGFBR1, and CDK1/2/9 were specific to an activated *Kras*[LSL-comQ61R] allele (*Figure 5*).

In summary, we suggest that tissues differ in their sensitivities to quantitative and/or qualitative RAS signaling both as a product of oncoprotein signaling and the cellular response thereof. We find that the level of oncogenic activity favors hematolymphopoietic neoplasias, the G12D mutant uniquely gives rise to oral and forestomach squamous tumors, while lung adenocarcinomas are sensitive to both mutation type and expression levels. The unique signaling dependencies of these

tissues may, in turn, be capitalized upon to identify new therapeutic opportunities to target early tumorigenesis, when the tumors are particularly vulnerable, perhaps either as an early intervention or as a preventative measure in high-risk populations.

# Materials and methods

**Key resources table**

| Reagent type (species) or resource | Designation | Source or reference | Identifiers | Additional information |
|---|---|---|---|---|
| Antibody | Anti-FLAG (mouse monoclonal) | Sigma | F1804 | WB (1:1000) |
| Antibody | Anti-KRAS (mouse monoclonal) | Santa Cruz Biotechnology | sc-30 | WB (1:500) |
| Antibody | Anti-βTubulin (mouse monoclonal) | Sigma | T5201 | WB (1:10,000) |
| Antibody | Anti-SV40 large T antigen (rabbit monoclonal) | Cell Signaling | 15729 | WB (1:1000) |
| Strain, strain background (*Escherichia coli*) | STBL3 | Thermo Fisher Scientific | C737303 | Chemically competent cells |
| Commercial assay or kit | FuGENE 6 Transfection Reagent | Promega | E2691 | |
| Commercial assay or kit | DC Protein Assay | Bio-Rad | 5000112 | |
| Commercial assay or kit | DNeasy Blood and Tissue DNA extraction Kit | QIAGEN | 69504 | |
| Chemical compound, drug | Tamoxifen | Sigma-Aldrich | T5648-5G | |
| Commercial assay or kit | RNAeasy Kit | QIAGEN | 74104 | |
| Commercial assay or kit | RNase-Free DNase Set | QIAGEN | 79254 | |
| Commercial assay or kit | GenPoint kit | Agilent | K062011-2 | |
| Commercial assay or kit | MycoAlert PLUS Mycoplasma detection kit | Lonza | LT07-703 | |
| Antibody | Anti-rabbit IgG antibody | Vector Laboratories | BA-1000-1.5 | RPPA |
| Antibody | Anti-mouse IgG antibody | Vector Laboratories | BA-9200-1.5 | RPPA |
| Commercial assay or kit | Active Ras Detection Kit | Cell Signaling | 8821 | |
| Commercial assay or kit | Ras GTPase ELISA Kit | Abcam | ab134640 | |
| Cell line (*Homo sapiens*) | HEK-HT | **Counter et al., 1992** | Cell line created and maintained in C. Counter lab, validated by immunoblot for SV40 Large T antigen | Cultured in DMEM and 10% FBS; Tested negative for mycoplasma |
| Cell line (*Mus musculus*) | MEF derived from *Kras$^{LSL-natG12D/+}$* mice | This paper | Validated by genotyping PCR | Cultured in DMEM and 10% FBS; Tested negative for mycoplasma |

*Continued on next page*

*Continued*

| Reagent type (species) or resource | Designation | Source or reference | Identifiers | Additional information |
|---|---|---|---|---|
| Cell line (*M. musculus*) | MEF derived from *Kras$^{LSL-natQ61R/+}$* mice | This paper | Validated by genotyping PCR | Cultured in DMEM and 10% FBS<br>Tested negative for mycoplasma |
| Cell line (*M. musculus*) | MEF derived from *Kras$^{LSL-comG12D/+}$* mice | This paper | Validated by genotyping PCR | Cultured in DMEM and 10% FBS<br>Tested negative for mycoplasma |
| Cell line (*M. musculus*) | MEF derived from *Kras$^{LSL-comQ61R/+}$* mice | This paper | Validated by genotyping PCR | Cultured in DMEM and 10% FBS<br>Tested negative for mycoplasma |
| Strain, strain background (*M. musculus*) | *Kras$^{LSL-natG12D/+}$* | This paper | | Generated in Counter Lab |
| Strain, strain background (*M. musculus*) | *Kras$^{LSL-natQ61R/+}$* | This paper | | Generated in Counter |
| Strain, strain background (*M. musculus*) | *Kras$^{LSL-comG12D/+}$* | This paper | | Generated in Counter Lab |
| Strain, strain background (*M. musculus*) | *Kras$^{LSL-comQ61R/+}$* | This paper | | Generated in Counter Lab |
| Strain, strain background (*M. musculus*) | *Kras$^{LSL-rareG12D/+}$* | This paper | | Generated in Counter Lab |
| Strain, strain background (*M. musculus*) | ACTB$^{FLPe/FLPe}$ | Jackson Laboratory | 003800 | |
| Strain, strain background (*M. musculus*) | *CC10$^{CreER/CreER}$; Rosa26$^{CAG-fGFP/CAG-fGFP}$* | *Xu et al., 2012* | | |
| Strain, strain background (*M. musculus*) | *Rosa26$^{CreERT2/CreERT2}$* | Jackson Laboratory | 008463 | |
| Recombinant DNA reagent | pcDNA3.1 | Thermo Fisher Scientific | V79020 | Mammalian expression vector backbone |
| Recombinant DNA reagent | pcDNA3.1+FLAG-Kras$^{natG12D}$ | This study | | Generated in Counter Lab |
| Recombinant DNA reagent | pcDNA3.1+FLAG-Kras$^{natQ61R}$ | This study | | Generated in Counter Lab |
| Recombinant DNA reagent | pcDNA3.1+FLAG-Kras$^{comG12D}$ | This study | | Generated in Counter Lab |
| Recombinant DNA reagent | pcDNA3.1+FLAG-Kras$^{comQ61R}$ | This study | | Generated in Counter Lab |
| Recombinant DNA reagent | pBABE-neo largeTcDNA | *Hahn et al., 2002* | Addgene #1780 | |
| Recombinant DNA reagent | MSCV-Cre-Hygro | *Wang et al., 2010* | Addgene #34565 | |
| Software, algorithm | ImageJ version 1.52k with Java 1.8.0_172 | *Schneider et al., 2012* | https://imagej.nih.gov/ij/ | |

Continued

| Reagent type (species) or resource | Designation | Source or reference | Identifiers | Additional information |
|---|---|---|---|---|
| Software, algorithm | Image Lab | Bio-Rad | https://www.bio-rad.com/en-us/product/image-lab-software | |
| Software, algorithm | GraphPad Prism v8 | GraphPad | https://www.graphpad.com/ | |
| Software, algorithm | bcl2fastq | Illumina | https://support.illumina.com/sequencing/sequencing_software/bcl2fastq-conversion-software.html | |
| Software, algorithm | STAR RNA-Seq alignment tool v2.7.8a | *Dobin et al., 2013* | https://github.com/alexdobin/STAR | STAR (RRID:SCR_004463) |
| Software, algorithm | DESeq2 | *Love et al., 2014* | http://www.bioconductor.org/packages/release/bioc/html/DESeq2.html | |
| Software, algorithm | GSEA | *Mootha et al., 2003* | https://www.gsea-msigdb.org/gsea/ | |
| Software, algorithm | R Studio | *R Development Core Team, 2020* | https://www.R-project.org | |
| Software, algorithm | MicroVigene Software Version 5.1.0.0 | VigeneTech | http://www.vigenetech.com/Protein.htm | |
| Software, algorithm | BioRender | | http://www.biorender.com | |

## Generation of *Kras*^LSL-natG12D, *Kras*^LSL-natQ61R, *Kras*^LSL-comG12D, and *Kras*^LSL-comQ61R alleles

A bacteria artificial chromosome was engineered with 7.5 kbp of 5′ flanking sequence *Kras* intron 1 DNA, a Lox-STOP-Lox cassette (*LSL*) (*Feil et al., 1996*), the first three coding exons fused together and encoded by either native (*nat*) codons or with 93 rare codons converted to the most commonly used codons in the mouse genome (*com*) and either a G12D or Q61 oncogenic mutation, followed by the N-terminal 564 bp of intron 4, an FRT-Neomycin-FRT cassette, and a further 1.5 kbp of 3′ flanking sequence (*Figure 1A*, *Figure 1—figure supplement 1*). The targeting cDNAs were each cloned into the targeting PL253 vector (*Liu et al., 2003*) and electroporated into 129S6/C57BL/6N (G4) ES cells. After selection, clones were screened using PCR and positive clones were selected for each engineered *Kras* allele, expanded, and frozen. At least two targeted clones per allele confirmed with a Southern hybridization were then microinjected into blastocysts to produce chimeras using standard procedures (*Behringer, 2014*). *Kras*^LSL-natG12D(+neo)/+, *Kras*^LSL-natQ61R(+neo)/+, *Kras*^LSL-comG12D(+neo)/+, and *Kras*^LSL-comQ61R(+neo)/+, chimeras (still retaining the *neo* cassette in the engineered *Kras* alleles) were crossed back to 129S6 mice. A genotyping PCR specific to the engineered *Kras* alleles with *neo* cassette was used to screen for germline transmission in clones. Each 129S6-*Kras*^LSL-nat(+neo)/+ and 129S6-*Kras*^LSL-com(+neo)/+ cohort was crossed with *ACTB*^FLPe/FLPe (Jackson Laboratory, strain 003800) mice to remove the selection marker via FLP-mediated excision of the *neo* cassette (*Dymecki, 1996*). Removal of the *neo* cassette was confirmed with genotyping PCR. Resultant strains were backcrossed with 129S6 mice for five generations, generating the *Kras*^LSL-natG12D/+, *Kras*^LSL-natQ61R/+, *Kras*^LSL-comG12D/+, and *Kras*^LSL-comQ61R/+ strains used in this study. All mouse care and experiments were performed in accordance with a protocol approved by the Institutional Animal Care and Use Committee (IACUC) of Duke University (protocol no. A195-19-09).

## Codon usage plots

The codon usage index (*Sharp and Li, 1987*) was calculated using the relative codon frequency derived from codon usage in the mouse exome (*Nakamura et al., 2000*) with a sliding windows of 25 codons across the open reading frame (ORF) of each transcript. A theoretical murine *Kras* ORF encoded by the rarest codons at each position (gray dotted line) was plotted for reference (*Figure 1—figure supplement 1*).

## Genotyping

Genomic DNA was isolated from 1 to 2 mm piece of toes by boiling for 30 min in 100 µl Toe Lysis Buffer (25 mM NaOH and 0.2 mM EDTA), followed by neutralization with 1.5 volume of Neutralization

Buffer (40 mM Tris–HCl pH 6.0). 1.2 µl of the isolated genomic DNA was subjected to PCR. Genotyping cell lines and mouse tissue was performed as above with 20 ng of genomic DNA isolated with a QIAGEN DNeasy Blood and Tissue DNA Extraction Kit (QIAGEN, #69504). All genotyping PCR reactions were performed using 0.4 U (0.08 µl) Platinum Taq polymerase (Invitrogen, #10342046) in 12.5 µl reaction volume with final concentration of 1.5–2 mM MgCl$_2$, 0.2 mM of each dNTP, and 0.5 µM of each primer. Full-length gels and replicates are provided (***Figure 1—source data 2***, ***Figure 4—source data 1***). Mice were genotyped using the following primers (also provided in Appendix 1):

*Kras*$^{LSL(+neo)}$ alleles:

>Kras.in3.F: *5'*-TTGGTGTACATCACTAGGCTTCA-*3'*
>Kras.in3.R: *5'*-TGGAAAGAGTAAAGTGTGGTGGT-*3'*
>Kras.neo.F: *5'*-GTGGGCTCTATGGCTTCTGA-*3'*
>Products: 590 bp (Targeted Allele[+neo]) or 240 bp (WT allele)

*Kras*$^{LSL-com}$ alleles:

>KrasCOM5.F: *5'*-CTTCCATTTGTCACGTCCTGC-*3'*
>KrasCOM5.R: *5'*-TCTTCGGTGGAAACAACGGT-*3'*
>Product: 448 bp (*Kras*$^{LSL-com}$)

*Kras*$^{LSL-nat}$ alleles:

>F-LSL: *5'*-TAGTCTGTGGGACCCCTTTG-*3'*
>R-LSL: *5'*-GCCTGAAGAACGAGATCAGC-*3'*
>Product: 448 bp (*Kras*$^{LSL-nat}$)

Recombination PCR:

>KRASOP.A2: *5'*-CTAGCCACCATGGCTTGAGT-*3'*
>KRASOP.B: *5'*-GTAATCACAACAAAGAGAATGCAG-*3'*
>LSL-F: *5'*-GGGGAACCTTTCAGGCTTA-*3'*
>Products: 616 bp (*Lox*P Recombined), 488 bp (WT allele), or 389 bp (*Kras*$^{LSL}$)

*CC10*$^{CreER}$ alleles:

>F-CC10 WT: *5'*-ACTCACTATTGGGGGTGTGG-*3'*
>R-CC10 WT: *5'*-GGAGGACTTGTGGATCTTG-*3'*
>F-Cre: *5'*-TCGATGCAACGAGTGATGAG-*3'*
>R-Cre: *5'*-TTCGGCTATAGGTAACAGGG-*3'*
>Products: 450 bp (*CC10*$^{CreER}$ allele) or 350 bp (WT allele)

*Rosa26*$^{CAG-fGFP}$ alleles:

>F-Rosa-01: *5'*-CACTTGCTCTTCCAAAGTCG-*3'*
>R-Rosa-02B: *5'*-TAGTCTAACTCGCGACACTG-*3'*
>F-CAG-02B: *5'*-GTTATGTAACGCGGAACTCC-*3'*
>Products: 500 bp (WT allele allele) or 350 bp (*Rosa26*$^{fGFP}$ allele)

Rosa26$^{CreERT2}$ alleles:

>R26R-univF: *5'*-AAAGTCGCTCTGAGTTGTTAT-*3'*
>R26R-wtR: *5'*-GGAGCGGGAGAAATGGATATG-*3'*
>CreER-R1: *5'*-CCTGATCCTGGCAATTTCG-*3'*
>Products: 800 bp (*Rosa26*$^{CreERT2}$ allele) or 600 bp (WT allele)

*Kras* LSL element:

>KRASOP.A2: *5'*-CTAGCCACCATGGCTTGAGT-*3'*
>KRASOP.B: *5'*-GTAATCACAACAAAGAGAATGCAG-*3'*
>Product: 389 bp (*Kras*$^{LSL}$ alleles)

CreER validation:

>F-Cre: *5'*-GGAGGACTTGTGGATCTTG-*3'*
>CREER-R1: *5'*-CCTGATCCTGGCAATTTCG-*3'*
>Product: 500 bp (CreER)

$Kras^{LSL-rare}$ alleles:

> KrasRAR.F: 5'-TATGCGTACGGGTGAAGGTT-3'
> KrasRAR.R: 5'-GCAGAGCACAGACTCACGTC-3'
> Product: 275 bp ($Kras^{LSL-rare}$)

## Plasmids

N-terminal FLAG-tagged murine *Kras* cDNAs encoding $Kras^{natG12D}$, $Kras^{natQ61R}$, $Kras^{comG12D}$, and $Kras^{comQ61R}$ were designed using the same sequence of the first three coding exons as described in the engineered alleles with the addition of the native *Kras4B* to the C-terminus and cloned into pcDNA3.1+ (Thermo Fisher). Plasmid sequences were verified by sequencing.

## Ectopic expression, immunoblots, and Ras activity assay

To validate the expression levels of the FLAG-tagged murine *Kras* cDNA constructs, $2 \times 10^6$ HEK-HT cells (*Counter et al., 1992*) were seeded in 10 cm tissue culture plates in DMEM with high glucose (Sigma-Aldrich, D5796) supplemented with 10% FBS (VWR, 97068-085), and transiently transfected the next day with the pcDNA3.1+ (empty vector) or the same plasmid encoding FLAG-$Kras^{natG12D}$, FLAG-$Kras^{natQ61R}$, FLAG-$Kras^{comG12D}$, or FLAG-$Kras^{comQ61R}$ using FuGene 6 reagent (Promega, E2691) according to the manufacturer's protocol. 48 hrs later, transfected cells were washed with cold PBS and pelleted. Cell pellets were lysed in 5 volumes of 1× lysis buffer (50 mM Tris–HCl pH 8.0, 150 mM NaCl, 1% NP40, 0.5% sodium deoxycholate, 0.1% SDS, and 5 mM EDTA) containing Protease Inhibitor Cocktail (Roche, 11836170001). The DC Protein Assay (Bio-Rad, 5000112) was used to measure protein concentration and 30 µg protein of each sample was resolved by SDS-PAGE, transferred to a PVDF membrane (Bio-Rad, 1704273), blocked in 5% milk, and immunoblotted by the following antibodies in 5% BSA (Sigma, A7906-500G): FLAG (Sigma, F1804; diluted 1:1000), KRAS (Santa Cruz, SC-30; diluted 1:500), and ß-Tubulin (Sigma, T5201; diluted 1:10,000). Primary antibody incubation was performed at room temperature for 1 hr followed by the secondary antibody incubation at 4°C overnight. To measure biochemical activity of the FLAG-tagged *Kras* constructs, cells were seeded, transfected, washed with PBS as above, after which cell pellets were lysed and immediately assayed with the Active Ras Detection Kit (Cell Signaling, #8821) or Ras GTPase ELISA Kit (Abcam, ab134640) according to the manufacturer's protocol. Measurements were done with two technical replicates of four serial dilutions and data shown are representative of two independent biological replicates. HEK-HT cells were confirmed to be mycoplasma-negative. Full-length images of immunoblots and replicates are provided (*Figure 1—source data 1*).

## Mouse embryonic fibroblasts (MEFs)

$Kras^{LSL-natG12D/+}$, $Kras^{LSL-natQ61R/+}$, $Kras^{LSL-comG12D/+}$, and $Kras^{LSL-comQ61R/+}$ mice were crossed with 129S6 mice for timed pregnancies to isolate embryos 12.5 to 13.5 days postcoitum, as previously described (*Serrano et al., 1997*). Pairs of primary MEF cultures derived from two separate embryos from each cohort were cultured in DMEM with high glucose (Sigma-Aldrich, D5796) supplemented with 10% FBS (VWR, 97068-085) and then immortalized by infection with an amphotropic retrovirus derived from plasmid pBABE-neo largeTcDNA (a gift from Robert Weinberg, Addgene plasmid #1780) (*Hahn et al., 2002*) using standard methodologies (*O'Hayer and Counter, 2006*). Paired parental MEF lines that were not infected with the pBABE-neo largeTcDNA were used as a negative control for antibiotic selection. Stable immortalized MEF cultures were selected in 400 µg/ml Geneticin (Thermo Fisher, 10131035) and again infected with an amphotrophc retrovirus derived from plasmid MSCV-Cre-Hygro (a gift from Kai Ge, Addgene plasmid #34565) to activate the mutant *Kras* allele (*Wang et al., 2010*). The stable Cre-expressing MEF cultures were selected in 500 µg/ml Hygromycin (Thermo Fisher, 10687010). Paired immortalized $Kras^{LSL-natG12D/+}$ MEF lines that were not infected with MSCV-Cre-Hygro were used as negative control for Hygromycin selection. Genomic DNA was isolated from the resultant cells using QIAGEN DNeasy Blood and Tissue DNA Extraction Kit (QIAGEN, #69504) and subjected to PCR to detect recombination of the $Kras^{LSL}$ alleles as described above. Paired immortalized MEF lines for each clone that were not infected with the MSCV-Cre-Hygro were used as negative control for recombination PCR. MEF cultures were confirmed to be mycoplasma-negative. Full-length gel image is provided (*Figure 1—source data 2*).

## Immunoblot and Ras activity assays in MEF cultures

The aforementioned pairs of independently derived $Kras^{LSL-natG12D/+}$, $Kras^{LSL-natQ61R/+}$, $Kras^{LSL-comG12D/+}$, and $Kras^{LSL-comQ61R/+}$ MEF cultures stably expressing Cre were grown in DMEM with high glucose (Sigma-Aldrich, D5796) supplemented with 10% FBS (VWR, 97068-085) in 15 cm dishes to prepare lysates for immunoblots and Ras activity assays. We note here that immortalized MEF cultures derived from $Kras^{LSL-comQ61R/+}$ mice exhibited decreased recombination rate compared to MEFs derived from the other three genotypes, potentially due to resistance to hygromycin. Given this, we generated single colonies from this line after hygromycin selection for all experiments investigating Kras expression and activity. Two independent cell lines from immortalized $Kras^{LSL-natG12D/+}$ MEF cultures without Cre were used as negative control to Cre-induced Kras expression. Once cells reached 85 to 90% confluency, FBS supplemented media was removed and replaced with plain DMEM with high glucose (Sigma-Aldrich, D5796) without FBS overnight. For serum stimulated lysates, cells were serum stimulated for 5 min by adding FBS to a final concentration of 10%. After serum starvation and stimulation, MEF cells were washed with cold PBS and pelleted. Cell pellets were lysed in 1 ml of lysis buffer provided in the Active Ras Detection Kit (Cell Signaling, #8821) supplemented with Halt Protease Inhibitor Cocktail (Thermo Fisher, 78430). The DC Protein Assay (Bio-Rad, 5000112) was used to measure protein concentration and 20 µg protein of each sample was used to immunoblot for total cell lysates. To measure biochemical activity of the endogenous Kras, 500 µg or 200 µg of each cell lysate was immediately assayed with the Active Ras Detection Kit (Cell Signaling, #8821) according to the manufacturer's protocol, except that incubation of lysates with RBD peptide was performed at 4°C overnight. Total cell lysates and lysates after RBD pull-down were resolved by 18% SDS-PAGE, transferred to a PVDF membrane (Bio-Rad, 1704273), blocked in 5% milk, and immunoblotted by the following antibodies in 5% BSA (Sigma, A7906-500G): KRAS (Santa Cruz, SC-30; diluted 1:500) and ß-Tubulin (Sigma, T5201; diluted 1:10,000). Primary antibody incubation was performed at 4°C overnight followed by the secondary antibody incubation for 3 hrs at room temperature. Immunoblots shown are representative of two independent MEF lines. Full-length images of immunoblots and replicates are provided (*Figure 1—source data 3*).

## Tumorigenesis studies

$Kras^{LSL-natG12D/+}$, $Kras^{LSL-natQ61R/+}$, $Kras^{LSL-comG12D/+}$, and $Kras^{LSL-comQ61R/+}$ mice were crossed with either $CC10^{CreER/CreER};Rosa26^{CAG-fGFP/CAG-fGFP}$ (a gift from Mark Onaitis) (*Xu et al., 2012*) or $Rosa26^{CreERT2/CreERT2}$ (Jackson Laboratory, strain 008463) mice and resultant offspring with the desired alleles were selected by genotyping. At six to eight weeks of age, selected littermates with random distribution of males and females received four intraperitoneal injections of tamoxifen (Sigma-Aldrich, T5648-5G, CAS# 10540-29-1) dissolved in corn oil (Sigma-Aldrich, C8267) and filter sterilized for a dose of 250 µg/g body weight to induce Cre-mediated recombination of the *LSL* cassette, activating the engineered oncogenic mutant *Kras* alleles. Mice from the first cross were humanely euthanized 1, 2, 3, 4, 5, or 6 months later while mice from the second cross were euthanized upon moribundity. To prevent age-related outcomes, four mice that did not reach moribundity by 300 days were euthanized at one year of age, which we reasoned to suffice to show tumorigenesis as it is about eight times longer than the previously identified average moribundity of five to six weeks with the tamoxifen-treated $Rosa26^{CreERT2/+};Kras^{LSL-G12D/+}$ mice (*Parikh et al., 2012*). Selected tissues were removed at necropsy and fixed in 10% formalin (VWR, 89370-094) for 24 to 48 hrs, then post-fixed in 70% ethanol (VWR, 89125-166) until analysis.

## Tissue analysis

Animals were euthanized with inhaled carbon dioxide and subjected to a complete necropsy. Selected organs were sampled for microscopic examination, including lung, liver, kidney, spleen, thymus, stomach, pancreas, and macroscopic lesions. All tissues were fixed for 48 hrs in 10% neutral buffered formalin (VWR, 89370-094) and then post-fixed in 70% ethanol (VWR, 89125-166), processed routinely, embedded in paraffin with the flat sides down, sectioned at a depth of 5 µm, and stained by the H&E method. Routine processing of the lungs from $CC10^{CreER/+};Rosa26^{CAG-fGFP/+}$ mice was performed by the Duke Research Immunohistology Lab, while all tissues from $Rosa26^{CreERT2/+}$ mice were processed by IDEXX Laboratories. Tissues and H&E slides were evaluated by a board-certified veterinary anatomic pathologist with experience in murine pathology (*Supplementary file 4 and 5*).

## Tissue recombination analysis

At six to eight weeks of age, two female mice from each of the genotypes $Kras^{LSL-natG12D/+}$, $Kras^{LSL-natQ61R/+}$, $Kras^{LSL-comG12D/+}$, and $Kras^{LSL-comQ61R/+}$ in a $Rosa26^{CreERT2/+}$ background were injected with tamoxifen as above, and seven days later euthanized. One age-matched female $Rosa26^{CreERT2/+};Kras^{LSL-comG12D/+}$ mouse was alsoeuthanized as a no-tamoxifen control. At necropsy, the colon, duodenum, ileum, cecum, jejunum, pancreas, spleen, glandular stomach, forestomach, kidney, liver, lung, heart, and ovaries were removed, genomic DNA extracted, and the status of recombination of each of the four oncogenic mutant $Kras^{LSL}$ alleles determined by recombination PCR in duplicate, as described above. The intensities of bands corresponding to the unaltered wild-type $Kras$ allele (WT) as well as the unrecombined and recombined oncogenic mutant $Kras^{LSL}$ alleles were quantified with ImageJ software and the recombination rates for each tissue type from each mouse calculated by dividing the densitometry of recombined allele to that of the wild-type allele. Full-length gels and replicates are provided (*Figure 4—source data 1*).

## Statistical analysis

Statistical analyses were performed using GraphPad Prism software version 8 (GraphPad Software). One-way ANOVA with Bonferroni's multiple-comparisons test with a single pooled variance and a 95% CI were used for experiments with more than two groups. Reported p-values are adjusted to account for multiple comparisons. A p-value of <0.05 was considered statistically significant.

## Ras activity assay on lung tissue

At six to eight weeks of age, two mice with random distribution of males and females from each of the genotypes $Rosa26^{CreERT2/+};Kras^{LSL-natG12D/+}$, $Rosa26^{CreERT2/+};Kras^{LSL-natQ61R/+}$, $Rosa26^{CreERT2/+};Kras^{LSL-comG12D/+}$, $Rosa26^{CreERT2/+};Kras^{LSL-comQ61R/+}$, and $Rosa26^{CreERT2/+}$ (control) were injected with tamoxifen as above, seven days latereuthanized, and their lungs were removed. Lungs were lysed in 5 ml of lysis buffer provided in the Active Ras Detection Kit (Cell Signaling, #8821) supplemented with Halt Protease Inhibitor Cocktail (Thermo Fisher, 78430). The DC Protein Assay (Bio-Rad, 5000112) was used to measure protein concentration, and 30 µg protein of each sample was used to immunoblot for total cell lysates. To measure biochemical activity of the endogenous Kras, 1 mg of each cell lysate was immediately assayed with the Active Ras Detection Kit (Cell Signaling, #8821) according to the manufacturer's protocol, except for overnight incubation of the RBD pull-down reaction at 4°C. 30 µg of total cell lysates and 20% of the RBD pull-down mixture were resolved by 8 to 18% gradient SDS-PAGE, transferred to a PVDF membrane (Bio-Rad, 1704273), blocked in 5% milk, and immunoblotted by the following antibodies in 5% BSA (Sigma, A7906-500G): KRAS (Santa Cruz, SC-30; diluted 1:500), and ß-Tubulin (Sigma, T5201; diluted 1:10,000). Primary antibody incubation was performed at 4°C overnight followed by the secondary antibody incubation for 2 hrs at room temperature. Full-length images of immunoblots and replicates are provided (*Figure 1—source data 4*).

## RNA-seq

For the first experiment, at six to eight weeks of age, three mice with random distribution of males and females from each of the genotypes $Rosa26^{CreERT2/+};Kras^{LSL-natG12D/+}$, $Rosa26^{CreERT2/+};Kras^{LSL-natQ61R/+}$, $Rosa26^{CreERT2/+};Kras^{LSL-comG12D/+}$, $Rosa26^{CreERT2/+};Kras^{LSL-comQ61R/+}$, and $Rosa26^{CreERT2/+}$ (control) were injected with tamoxifen as above, seven days later euthanized, and their lungs were removed. For the second RNA-seq experiment, the experiment was repeated exactly as before except with $Rosa26^{CreERT2/+};Kras^{LSL-natG12D/+}$ versus $Rosa26^{CreERT2/+};Kras^{LSL-comQ61R/+}$ mice. Tissue lysis and RNA extraction steps were performed in a chemical hood, and all instruments and tools were sprayed with RNaseZAP (Sigma, R2020) to prevent RNA degradation. Isolated lungs were immediately stored in RNA stabilizing solution RNAlater (Sigma, R0901) at 4°C overnight and then transferred to –80°C for long-term storage. Lung tissues were thawed, weighed, and pulverized with mortar and pestle in the presence of liquid nitrogen. RNA extraction was performed immediately thereafter using RNAeasy Kit (QIAGEN, 74104) according to the manufacturer's instructions. Briefly, tissue lysates were prepared according to the kit instructions and tissue clumps were removed using QIAshredder columns (QIAGEN, 79654). Cleared lysates were applied to RNeasy silica columns and on-column DNase digestion was performed using RNase-Free DNase Set (QIAGEN, 79254) to remove DNA in silica membrane. Following wash steps, RNA was eluted in RNase-free water that was treated with RNAsecure RNase Inactivation Reagent

(Thermo Fisher, AM7005) according to the manufacturer's instructions. Extracted total RNA quality and concentration were assessed on Fragment Analyzer (Agilent Technologies) and Qubit 2.0 (Thermo Fisher Scientific), respectively. Samples with RIN less than 7 were not sequenced. RNA-seq libraries were prepared using the commercially available KAPA Stranded mRNA-Seq Kit (Roche). In brief, mRNA transcripts were captured from 500 ng of total RNA using magnetic oligo-dT bead. The mRNA was then fragmented using heat and magnesium, and reverse transcribed using random priming. During second-strand synthesis, the cDNA:RNA hybrid was converted into to double-stranded cDNA (dscDNA) and dUTP was incorporated into the second cDNA strand, effectively marking this strand. Illumina sequencing adapters were ligated to the dscDNA fragments and amplified to produce the final RNA-seq library. The strand marked with dUTP was not amplified, allowing for strand-specific sequencing. Libraries were indexed using a dual-indexing approach allowing for all the libraries to be pooled and sequenced on the same sequencing run. Before pooling and sequencing, fragment length distribution for each library was first assessed on a Fragment Analyzer (Agilent Technologies). Libraries were also quantified using Qubit. Molarity of each library was calculated based on Qubit concentration and average library size. All libraries were then pooled in equimolar ratio and sequenced. Sequencing was done on an Illumina NovaSeq 6000 sequencer. The pooled libraries were sequenced on one lane of an S-Prime flow cell at 50 bp paired-end. Once generated, sequence data was demultiplexed and Fastq files generated using bcl2fastq v2.20.0.422 file converter from Illumina. The RNA-seq data has been deposited in NCBI's Gene Expression Omnibus (*Edgar et al., 2002*). Initial and secondary RNA-seq data are available through GEO Series accession numbers GSE181628 and GSE181627, respectively, and are accessible under the project GSE181629 (https://www.ncbi.nlm.nih.gov/geo/query/acc.cgi?acc=GSE181629).

## Transcriptome analysis

RNA-seq data was processed using the TrimGalore toolkit (http://www.bioinformatics.babraham.ac.uk/projects/trim_galore), which employs Cutadapt (*Martin, 2011*) to trim low-quality bases and Illumina sequencing adapters from the 3′ end of the reads. Only reads that were 20 nucleotides or longer after trimming were kept for further analysis. Reads were mapped to the GRCm38v73 version of the mouse genome and transcriptome (*Kersey et al., 2012*) using the STAR RNA-seq alignment tool (*Dobin et al., 2013*). Reads were kept for subsequent analysis if they mapped to a single genomic location. Gene counts were compiled using the HTSeq tool (http://www-huber.embl.de/users/anders/HTSeq/). Only genes that had at least ten reads in any given library were used in subsequent analysis. Normalization and differential expression were carried out using the DESeq2 (*Love et al., 2014*) Bioconductor (*Huber et al., 2015*) package with the R statistical programming environment (http://www.R-project.org). The false-discovery rate was calculated to control for multiple hypothesis testing (GSE181628). GSEA (*Mootha et al., 2003*) was performed to identify hallmarks and pathways associated with altered gene expression for each of the comparisons performed (*Figure 3B and C*, *Figure 3—figure supplement 8*, *Figure 3—source data 2*). To identify transcription factors significantly enriched, the differential expression data from $Kras^{LSL-natG12D/+}$ and $Kras^{LSL-comQ61R/+}$ transcriptomes was trimmed to genes whose adjusted p-value is less than 5% and whose $\log_2$ fold-change either larger than +1 or lower than –1. The trimmed dataset was cross-referenced with TRANSFAC, a curated eukaryotic transcription factor dataset, retrieved from Harmonizome web portal (*Matys et al., 2003*; *Matys et al., 2006*; *Rouillard et al., 2016*). Retrieved gene transcription factor pairs from TRANSFAC were then censored to the transcription factors enriched in $Kras^{LSL-natG12D/+}$ and $Kras^{LSL-comQ61R/+}$ specific GSEA hallmarks including more than four such gene transcription factor pairs both with p-value less than 5% (*Figure 5—figure supplement 1*). To identify low and high Kras-specific druggable kinases, the transcriptome enriched in $Kras^{LSL-natG12D/+}$ and $Kras^{LSL-comQ61R/+}$ specific GSEA hallmarks were cross-referenced with kinases whose clinical inhibitors were previously surveyed (*Klaeger et al., 2017*). The kinases enriched in $Kras^{LSL-natG12D/+}$ and $Kras^{LSL-comQ61R/+}$ are shown in blue and red, respectively, with adjusted p-value less than 5% highlighted (*Figure 5*).

## Reverse transcription quantitative real-time PCR (qRT-PCR)

At six to eight weeks of age, two to three mice with random distribution of males and females from each of the genotypes $Rosa26^{CreERT2/+};Kras^{LSL-natG12D/+}$, $Rosa26^{CreERT2/+};Kras^{LSL-natQ61R/+}$, $Rosa26^{CreERT2/+};Kras^{LSL-comG12D/+}$, $Rosa26^{CreERT2/+};Kras^{LSL-comQ61R/+}$, and $Rosa26^{CreERT2/+}$ (control) were injected with tamoxifen as

above, and seven days later euthanized to harvest lungs. RNA extraction was performed as mentioned above, followed by first-strand cDNA synthesis from 2 µg RNA, and real-time quantitative PCR using GoTaq 2-Step RT-qPCR kit (Promega, A6110). All measurements were normalized against Actin as the internal control using the $2^{-\Delta\Delta Ct}$ method (*Figure 1—source data 5*, *Figure 3—source data 3*). For qRT-PCR analysis for the second biological replicate, one of two negative controls did not have numerical data for two transcripts and hence was not used for plotting. Data shown are representative of two independent biological replicates, and the primer sequences are provided in Appendix 1.

## Reverse-phase protein microarray (RPPA)

At six to eight weeks of age, three to four mice with random distribution of males and females from each of the genotypes *Rosa26*$^{CreERT2/+}$;*Kras*$^{LSL-natG12D/+}$, *Rosa26*$^{CreERT2/+}$;*Kras*$^{LSL-comQ61R/+}$, and *Rosa26*$^{CreERT2/+}$ (control) were injected with tamoxifen as above, seven days later euthanized, and their lungs were removed. Lungs were snap froze, and embedded in Optimal Cutting Temperature (O.C.T.) compound (Tissue-Tek, 4583). Five sections of 8 µm were cut, mounted on uncoated glass slides, and stored at −80°C until lysed. Whole-tissue lysates were obtained directly from the glass slides after sections were fixed in 70% ethanol, washed with deionized water, stained with hematoxylin (Sigma-Aldrich, H9627, CAS# 517-28-2) and Scott's Tap Water (Electron Microscopy Sciences, 2607006), and dehydrated in an ascending series of ethanol (70, 95, and 100%). Solutions were supplemented with cOmplete Mini Protease Inhibitors (Roche Applied Science, 11836153001) to avoid proteins and phosphorylated residues degradation. Stained tissues were lysed in a 1:1 solution of Tissue Protein Extraction Reagent (Pierce, 78510) and 2× Tris-Glycine SDS Sample buffer (Invitrogen Life Technologies, LC2676) supplemented with 2.5% β-mercaptoethanol (Sigma-Aldrich, M3148, CAS# 60-24-2). Tissue lysates were immobilized onto nitrocellulose-coated slides (Grace Bio-labs, 505278) in three technical replicates using an Aushon 2470 arrayer (Quanterix) equipped with 185 µm pins as previously described (*Baldelli et al., 2017*). Reference standard curves for internal quality control were arrayed along with the tissue lysates along with a bovine serum albumin serial dilution curve to quantify total protein amount on each sample. To assess the amount of protein in each sample and for normalization purposes, selected arrays were stained with Sypro Ruby Protein Blot Stain (Molecular Probes, S12000) following the manufacturer's instructions as previously described (*Baldelli et al., 2021*). Reblot Antibody Stripping solution (Chemicon, 2500) was used to treat the remaining arrays for 15 min at room temperature before antibody staining. Arrays were subsequently washed with PBS and incubated for at least 4 hrs in I-block reagent (Tropix, T2015). Using an automated system (Dako Cytomation), arrays were probed with selected antibodies targeting total proteins and post-translationally modified epitopes of kinases and their downstream substrates (*Supplementary file 1*). For their use on the array, antibodies were validated as previously described (*Signore et al., 2017*). After incubation with a primary antibody, each array was probed with a biotinylated anti-rabbit (Vector Laboratories, Inc BA-1000-1.5) or anti-mouse (Vector Laboratories, BA-9200-1.5) secondary antibodies matching the species of the primary antibody. Signal detection was performed using the GenPoint kit (Agilent, K062011-2), a commercially available tyramide-based avidin/biotin amplification system, coupled with a fluorescent streptavidin-conjugated IRDye680 dye (LI-COR Biosciences, 680RD) according to the manufacturer's recommendation. As negative controls for the background signal, selected arrays were probed with the secondary antibody alone. Antibody, negative control, and Sypro Ruby-stained slides were scanned at the appropriate wavelength channel using a laser scanner (TECAN). Image analysis was performed by spot finding, subtraction of local background and unspecific binding generated by the secondary antibody, and normalization to the amount of total protein obtained from the Sypro Ruby-stained slide by MicroVigene Software Version 5.1.0.0 (VigeneTech) as previously described (*Pin et al., 2014*). Replicate values are averaged to generate a single-intensity RPPA value per sample. Raw data with these average values are shown in *Supplementary file 1*.

## RPPA analysis

RPPA data for each tissue was first processed by replacing intensity values of 0 with 1 to ease computation. Each value was then converted to $\log_2$ scale. To calculate $\log_2$ fold-change, we calculated the average $\log_2$ values of each target in each genotype and subtracted the average $\log_2$ values of the wild-type *Kras* allele of that target. Heatmaps for the $\log_2$ fold-change were plotted using R statistical programming environment (http://www.R-project.org). Note that RPPA analysis of one of the

*Rosa26^{CreERT2/+};Kras^{LSL-natG12D/+}* mouse lungs was deemed an outlier as it shared little commonality with the other three biological replicates. As such, the RPPA analysis of this mouse was not included in *Figure 1—figure supplement 6* and *Figure 3—figure supplement 9*.

## Generation of *Kras^{LSL-rareG12D/+}* mice and tumorigenesis study

An additional inducible *Kras* allele with the rarest codons was generated in the same manner as mentioned above from BAC design to chimera production (*Figure 1—figure supplement 1*). *Kras^{LSL-rareG12D(+neo)}* chimeras were crossed back to 129S6 mice followed by *ACTB^{FLPe/FLPe}* (Jackson Laboratory, strain 003800) mice to remove the *neo* selection marker via FLP-mediated excision (*Dymecki, 1996*). Both germline transmission and the removal of the *neo* cassette were confirmed with genotyping PCR as mentioned above. Resultant strain was backcrossed with 129S6 mice for five generations, generating the *Kras^{LSL-rareG12D/+}* strain. *Kras^{LSL-rareG12D/+}* mice were crossed with *Rosa26^{CreERT2/CreERT2}* (Jackson Laboratory, strain 008463) mice and animals with the desired alleles selected by genotyping. At six to eight weeks of age, seven mice with random distribution of males and females were injected with tamoxifen as above, and 22 months later euthanized. Lung, liver, kidney, spleen, thymus, stomach, pancreas, and femurs were removed at necropsy and fixed in 10% formalin (VWR, 89370-094) for 24 to 48 hrs, then post-fixed in 70% ethanol (VWR, 89125–166), processed routinely, embedded in paraffin with the flat sides down, sectioned at 5 µm, and stained with H&E. Routine processing of the tissues was performed by IDEXX Laboratories. Tissues and H&E slides were evaluated by a board-certified veterinary anatomic pathologist with experience in murine pathology.

## Acknowledgements

We thank members of the Counter Laboratory for technical assistance, David Kirsch, James Alvarez, Mark Onaitis, Seth Zimmerman, Channing Der, and Aaron Hobbs for advice, Cheryl Block, Gary Kucera, and the Duke Cancer Institute Transgenic Mouse Facility for the generation of mice, and Nicolas Devos, David Corcoran, Wei Chen, and Duke Center for Genomic and Computational Biology for their assistance in RNA-Seq. All mouse and organ pictures were created with Biorender.com. This work was supported by the National Cancer Institute (R01CA94184, R01CA269272, and P01CA203657 to CMC) and the Shared Resources of the Duke Cancer Institute (P30CA014236). The authors declare no competing interests.

## Additional information

### Competing interests

Elisa Baldelli, Mariaelena Pierobon, Emanuel F Petricoin: EB, MP, and EP are inventors on US Government and University assigned patents and patent applications that cover aspects of the technologies discussed such as the Reverse Phase Protein Microarrays. As inventors, they are entitled to receive royalties as provided by US Law and George Mason University policy. MP and EP receive royalties from and are consultants of TheraLink Technologies, Inc EP is shareholders and consultants of TheraLink Technologies, Inc EP is shareholder and consultant of Perthera, Inc. Christopher M Counter: co-founder of the company Merlon Inc. The other authors declare that no competing interests exist.

### Funding

| Funder | Grant reference number | Author |
|---|---|---|
| National Cancer Institute | R01CA94184 | Christopher M Counter |
| National Cancer Institute | R01CA269272 | Christopher M Counter |
| National Cancer Institute | P01CA203657 | Christopher M Counter |
| Duke Cancer Institute | | Christopher M Counter |

The funders had no role in study design, data collection and interpretation, or the decision to submit the work for publication.

### Author contributions

Özgün Le Roux, Conceptualization, Resources, Data curation, Formal analysis, Validation, Investigation, Visualization, Methodology, Writing – original draft, Writing – review and editing; Nicole LK Pershing, Conceptualization, Writing – review and editing; Erin Kaltenbrun, Conceptualization, Resources, Writing – review and editing; Nicole J Newman, Investigation, Writing – review and editing; Jeffrey I Everitt, Data curation, Methodology, Writing – review and editing; Elisa Baldelli, Mariaelena Pierobon, Data curation, Formal analysis, Methodology; Emanuel F Petricoin, Resources, Supervision; Christopher M Counter, Conceptualization, Resources, Supervision, Funding acquisition, Visualization, Writing – original draft, Project administration, Writing – review and editing

### Author ORCIDs

Özgün Le Roux  http://orcid.org/0000-0002-9885-6872
Jeffrey I Everitt  http://orcid.org/0000-0003-0273-6284
Christopher M Counter  http://orcid.org/0000-0003-0748-3079

### Ethics

All mouse care and experiments were performed in accordance with a protocol approved by the Institutional Animal Care and Use Committee (IACUC) of Duke University (Protocol no. A195-19-09).

### Decision letter and Author response

Decision letter https://doi.org/10.7554/eLife.75715.sa1
Author response https://doi.org/10.7554/eLife.75715.sa2

## Additional files

### Supplementary files

• Supplementary file 1. Raw RPPA data from the lungs of mice upon activating the $Kras^{LSL-natG12D}$ versus $Kras^{LSL-comQ61R}$ allele with respect to the wild-type $Kras$ allele.

• Supplementary file 2. Gel densitometer results of recombination analysis of 14 organs after tamoxifen injection in *Figure 4—figure supplement 2*.

• Supplementary file 3. Statistical analysis of pairwise comparisons of the survival plots upon activating the four inducible $Kras^{LSL}$ alleles in *Figure 4A*.

• Supplementary file 4. Histopathology report for H&E-stained tissues from multiple organs from each of the four $Kras^{LSL}$ alleles.

• Supplementary file 5. Grading of observed tumors in multiple organs from each of the four inducible $Kras^{LSL}$ alleles.

• Supplementary file 6. Transcription factors identified in Gene Set Enrichment Analysis (GSEA) hallmarks of lung transcriptome in *Figure 5—figure supplement 1*.

• MDAR checklist

### Data availability

Initial and secondary RNAseq data are available through GEO Series accession numbers GSE181628 and GSE181627, respectively, and are accessible under the project GSE181629 (https://www.ncbi.nlm.nih.gov/geo/query/acc.cgi?acc=GSE181629).

The following datasets were generated:

| Author(s) | Year | Dataset title | Dataset URL | Database and Identifier |
|---|---|---|---|---|
| Erdogan O, Counter CM | 2021 | Signaling response to early oncogenesis by a novel panel of murine mutant Kras alleles in the mouse lung transcriptome | https://www.ncbi.nlm.nih.gov/geo/query/acc.cgi?acc=GSE181628 | NCBI Gene Expression Omnibus, GSE181628 |
| Erdogan O, Counter CM | 2021 | Signaling response to early oncogenesis by a novel panel of murine mutant Kras alleles in the mouse lung transcriptome | https://www.ncbi.nlm.nih.gov/geo/query/acc.cgi?acc=GSE181627 | NCBI Gene Expression Omnibus, GSE181627 |

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

# Appendix 1

Primer sequences for genotyping and qRT-PCR analysis.

**Appendix 1—table 1.** Primer sequences for genotyping and qRT-PCR analysis.

| Reagent type (species) or resource | Designation | Source or reference | Identifiers | Additional information |
|---|---|---|---|---|
| Sequence-based reagent | Kras.in3.F | This paper | PCR primers | TTGGTGTACATCACTAGGCTTCA |
| Sequence-based reagent | Kras.in3.R | This paper | PCR primers | TGGAAAGAGTAAAGTGTGGTGGT |
| Sequence-based reagent | Kras.neo.F | This paper | PCR primers | GTGGGCTCTATGGCTTCTGA |
| Sequence-based reagent | KrasCOM5.F | This paper | PCR primers | CTTCCATTTGTCACGTCCTGC |
| Sequence-based reagent | KrasCOM5.R | This paper | PCR primers | TCTTCGGTGGAAACAACGGT |
| Sequence-based reagent | F-LSL | This paper | PCR primers | TAGTCTGTGGGACCCCTTTG |
| Sequence-based reagent | R-LSL | This paper | PCR primers | GCCTGAAGAACGAGATCAGC |
| Sequence-based reagent | KRASOP.A2 | This paper | PCR primers | CTAGCCACCATGGCTTGAGT |
| Sequence-based reagent | KRASOP.B | This paper | PCR primers | GTAATCACAACAAAGAGAATGCAG |
| Sequence-based reagent | LSL-F | This paper | PCR primers | GGGGAACCTTTCAGGCTTA |
| Sequence-based reagent | KrasRAR.F | This paper | PCR primers | TATGCGTACGGGTGAAGGTT |
| Sequence-based reagent | KrasRAR.R | This paper | PCR primers | GCAGAGCACAGACTCACGTC |
| Sequence-based reagent | F-CC10 WT | *Xu et al., 2012* | PCR primers | ACTCACTATTGGGGGTGTGG |
| Sequence-based reagent | R-CC10 WT | *Xu et al., 2012* | PCR primers | GGAGGACTTGTGGATCTTG |
| Sequence-based reagent | F-Cre | *Xu et al., 2012* | PCR primers | TCGATGCAACGAGTGATGAG |
| Sequence-based reagent | R-Cre | *Xu et al., 2012* | PCR primers | TTCGGCTATAGGTAACAGGG |
| Sequence-based reagent | F-Rosa-01 | *Xu et al., 2012* | PCR primers | CACTTGCTCTTCCAAAGTCG |
| Sequence-based reagent | R-Rosa-02B | *Xu et al., 2012* | PCR primers | TAGTCTAACTCGCGACACTG |
| Sequence-based reagent | F-CAG-02B | *Xu et al., 2012* | PCR primers | GTTATGTAACGCGGAACTCC |
| Sequence-based reagent | R26R-univF | Jackson Laboratory | PCR primers | AAAGTCGCTCTGAGTTGTTAT |
| Sequence-based reagent | R26R-wtR | Jackson Laboratory | PCR primers | GGAGCGGGAGAAATGGATATG |
| Sequence-based reagent | CreER-R1 | Jackson Laboratory | PCR primers | CCTGATCCTGGCAATTTCG |
| Sequence-based reagent | GLI1-F | This paper | PCR primers | CCCATAGGGTCTCGGGGTCTCAAAC |
| Sequence-based reagent | GLI1-R | This paper | PCR primers | GGAGGACCTGCGGCTGACTGTGTAA |
| Sequence-based reagent | CDH1-F | This paper | PCR primers | GTCTCCTCATGGCTTTGC |
| Sequence-based reagent | CDH1-R | This paper | PCR primers | CTTTAGATGCCGCTTCAC |
| Sequence-based reagent | TWIST1-F | This paper | PCR primers | AGCGGGTCATGGCTAACG |
| Sequence-based reagent | TWIST1-R | This paper | PCR primers | GGACCTGGTACAGGAAGTCGA |
| Sequence-based reagent | ZEB2-F | This paper | PCR primers | GAGCTTGACCACCGACTC |
| Sequence-based reagent | ZEB2-R | This paper | PCR primers | TTGCAGGACTGCCTTGAT |
| Sequence-based reagent | SOX5-F | This paper | PCR primers | ATTGTGCAGTCCCACAGGTTG |
| Sequence-based reagent | SOX5-R | This paper | PCR primers | CTGCCTTTAGTGGGCCAGTG |
| Sequence-based reagent | IL6-F | This paper | PCR primers | CCGGAGAGGAGACTTCACAG |
| Sequence-based reagent | IL6-R | This paper | PCR primers | CAGAATTGCCATTGCACAAC |
| Sequence-based reagent | IL10-F | This paper | PCR primers | GGTTGCCAAGCCTTATCGGA |
| Sequence-based reagent | IL10-R | This paper | PCR primers | ACCTGCTCCACTGCCTTGCT |

*Appendix 1—table 1 Continued on next page*

*Appendix 1—table 1 Continued*

| Reagent type (species) or resource | Designation | Source or reference | Identifiers | Additional information |
|---|---|---|---|---|
| Sequence-based reagent | TNFa-F | This paper | PCR primers | CCCCAAAGGGATGAGAAGTT |
| Sequence-based reagent | TNFa-R | This paper | PCR primers | GTGGGTGAGGAGCACGTAGT |
| Sequence-based reagent | CCL2-F | This paper | PCR primers | AGGTCCCTGTCATGCTTCTG |
| Sequence-based reagent | CCL2-R | This paper | PCR primers | TCTGGACCCATTCCTTCTTG |
| Sequence-based reagent | LIF-F | This paper | PCR primers | AATGCCACCTGTGCCATACG |
| Sequence-based reagent | LIF-R | This paper | PCR primers | CAACTTGGTCTTCTCTGTCCCG |
| Sequence-based reagent | TNFSF9-F | This paper | PCR primers | GCAAGCAAAGCCTCAGGTAG |
| Sequence-based reagent | TNFSF9-R | This paper | PCR primers | TCCAGGAACGGTCCACTAAC |
| Sequence-based reagent | IFNg-F | This paper | PCR primers | CGGGAGGTGCTGCTGATGG |
| Sequence-based reagent | IFNg-R | This paper | PCR primers | AGGGACAGCCTGTTACTACC |
| Sequence-based reagent | FOS-F | This paper | PCR primers | CCTGCCCCTTCTCAACGAC |
| Sequence-based reagent | FOS-R | This paper | PCR primers | GCTCCACGTTGCTGATGCT |
| Sequence-based reagent | FOSL2-F | This paper | PCR primers | GAGTCCTACTCCAGCGGTG |
| Sequence-based reagent | FOSL2-R | This paper | PCR primers | GACTGTAGGGATGTGAGCGT |
| Sequence-based reagent | DUSP6-F | *Li and Counter, 2021* | PCR primers | ACTTGGACGTGTTGGAAGAGT |
| Sequence-based reagent | DUSP6-R | *Li and Counter, 2021* | PCR primers | GCCTCGGGCTTCATCTATGAA |
| Sequence-based reagent | EGR1-F | *Li and Counter, 2021* | PCR primers | CCTGACCACAGAGTCCTTTTCT |
| Sequence-based reagent | EGR1-R | *Li and Counter, 2021* | PCR primers | AGGCCACTGACTAGGCTGA |
| Sequence-based reagent | FOSL1-F | *Li and Counter, 2021* | PCR primers | CAGGAGTCATACGAGCCCTAG |
| Sequence-based reagent | FOSL1-R | *Li and Counter, 2021* | PCR primers | GCCTGCAGGAAGTCTGTCAG |
| Sequence-based reagent | ACTIN-F | *Li and Counter, 2021* | PCR primers | CGTGAAAAGATGACCCAGATCATGT |
| Sequence-based reagent | ACTIN-R | *Li and Counter, 2021* | PCR primers | CGTGAGGGAGAGCATAGCC |

