## [Editor Report]

This article addresses the long-standing question of why specific mutations in RAS are associated with tumors arising in distinct tissues; whether this reflects biological expression or functionality. The authors utilize a clever mouse genetic approach to modulate both which Kras mutation is expressed and how much of that Kras mutant is expressed, expressing the same Kras mutation at either a low level (due to rare codon utilization) or at a higher level (through common codon utilization). This study convincingly supports the conclusion that different KRAS mutations impart unique signaling properties that are preferentially capable of inducing tumor initiation in a distinct cell-specific manner.

---

## [Decision Letter]

**Decision letter after peer review:**

Thank you for submitting your article "RAS mutation patterns arise from tissue-specific responses to distinct oncogenic signaling" for consideration by *eLife*. Your article has been reviewed by 3 peer reviewers, and the evaluation has been overseen by a Reviewing Editor and Erica Golemis as the Senior Editor. The following individuals involved in review of your submission have agreed to reveal their identity: Jonathan Chernoff (Reviewer #1); Aaron Hobbs (Reviewer #3).

Essential revisions:

1) Based on their discussion, the reviewers are in agreement that use of some method is required to demonstrate that the RNA changes documented in the study result in real changes in protein-level expression of key signaling molecules. The reviewers indicate this could be addressed using various approaches, including 1) total proteome analysis; 2) immunoblots for proteins that are important to the analysis; or 3) RPPA. However, some method of evaluating protein changes must be applied, and the results integrated into the analysis.

2) In addition, each of the three reviewers made some suggestions regarding phrasing and interpretation. The authors are strongly encouraged to address these.

*Reviewer #1 (Recommendations for the authors):*

The authors try to establish biological differences between commonly encountered KRAS mutants. Distinct mutations exhibit tissue tropism (e.g., KRAS G12C in lung, G12D and G12R in pancreas, etc), but the reasons for these distributions have never been fully clarified. By establishing mouse models in which the type of KRAS mutant and its level of expression can be readily controlled, the authors hope to establish whether the type of amount of KRAS mutant expressed will result in distinct outcomes.

Major strengths include the extensive body of work from the Counter lab regarding the key role of codon usage in KRAS oncogenicity, the generation of several sets of matched mouse models (two of which are studied in detail, but the others are also of value), and the examination of KRAS effects in many tissues.

Weaknesses include the fact (acknowledged by the authors) that the system is highly artificial, in that KRAS is being induced globally at one point in time, that its expression levels are artificially boosted by codon changes that would not occur in vivo (though perhaps might happen if the KRAS gene were amplified, as sometimes happens), and a lack of any protein level data to corroborate the RNA profiling performed in the study.

It would strengthen the work if the authors would back up some of the key RNA signatures with western blots of RPPA-type data.

It could be instructive if the authors performed exon sequencing on some of their tumors. They saw no pancreatic tumors, which is a bit surprising, though it was perhaps because tp53 was not also mutated. One wonder what mutations might have occurred in those tissues where tumors were seen, such as lung.

*Reviewer #2 (Recommendations for the authors):*

Within this manuscript, the authors utilize mouse genetic approaches to modulate both which Kras mutation is expressed and how much of that Kras mutant is expressed. By utilizing different codons, they can have the same Kras mutation expressed at either a low level (due to rare codon utilization) or at a higher level (through common codon utilization). This is a clever strategy that this group pioneered for studying RAS signaling, and by now comparing two different Kras mutants (G12D and Q61R) they can also introduce additional quantitative and qualitative variation.

The authors define quantitative variation as variation in the intensity of active Kras-GTP, and qualitative variation as how the Kras-GTP may activate different downstream effector proteins. This is a well-defined concept that helps provide a structure to their analysis.

The genetic approach, and the setting of definitions, are major strengths as the authors dissect this complicated problem in a logical, systematic manner. Additional strengths include studying how the different Kras strains differ for different tissue phenotypes.

However, there are a few weaknesses to the manuscript in its current form. Although the authors' hypotheses about quantitative and qualitative variation are likely true, at times the authors appear to interpret their data more strongly than the data seem to permit. In many of these cases, a rephrasing that describes their findings objectively would be valuable because the data – at minimum – highlight the need for these quantitative and qualitative factors to be better considered within the field.

One seemingly striking weakness is that the allelic series does not seem to be sufficiently well demonstrated to have the progressive increase in mutant strengths. Another, and partially overlapping weakness, is that the authors seem to invoke tissue-specific differences when convenient and to ignore them when convenient.

The field of RAS biology has made great progress ignoring the potential effects of signal intensity and mutant-to-mutant variation, but the reinvigoration of the field in recent years has revealed important mutant-to-mutant specific differences and instances where intensity matters. A deeper understanding of these factors, and improved tools to study these differences, are both important to the field. The authors are here advancing tools and approaches that could be used in future studies and are providing a foundation for future studies. Thus, the work is likely to be impactful and influential.

One concern is the demonstration of the progressive quantitative strength of the allelic series. Even though a mouse genetic approach is utilized for the manuscript, the comparisons of Kras mutant strength is made with transiently transfected HEK-HT cells. Could the authors not do this in the MEFs they utilize, which would be endogenous expression from these cells derived from the genetically engineered mice?

The authors later claim, and acknowledge, tissue differences in factors that can contribute to the overall activation of Kras, and the authors also acknowledge qualitative differences. It seems reasonable to consider that there could be tissue-specific differences in which GEFs are activated, and that active GEFs might act upon the G12D and Q61R mutants differently – just as effector proteins may bind the mutants differently. The authors seem to be assuming the relative strengths observed are invariant between cell/tissue types and that their observations are either reflective of these quantitative differences, or that if they differ, that they must be qualitative. But why can't there be qualitative upstream differences that impact the quantity of Kras-GTP? In other words – quantitative and qualitative influences still seem to be intertwined in this system. The authors may want to state that these are strong (and unproven) assumptions, and that their interpretations are based upon these assumptions.

The authors refer to "normal cells" in the introduction; they appear to be referring to HEK-HT cells. It seems misleading to refer to them as "normal". The authors should be more precise.

The title seems to be too strongly stated. Mutation patterns also arise from the mutagens (cigarette smoking in lung causing G12C, etc.). Additionally, the authors do not show this or prove this – this is a speculation down the road from their findings, by my interpretation of their study. A more objective title would be more appropriate.

The authors state "determining the immediate response of normal cells to different Kras mutations in vivo thus holds the key to understanding the mutational patterning of this oncogene". The logic to make such a strong conclusion is not apparent. This rather seems to be a hypothesis, that studying the immediate response may offer some insight into mutational patterning. Several specific details in the sentence appear overly strong. Can this be rephrased?

The sentence "In this way, tissues sensitive to quantitative signaling…". This is important, as it highlights the authors interpretation of their data (and possible flaws with this assumption are mentioned above). Could the authors rephrase this a bit more mildly to describe that the authors interpret their data to indicate quantitative signaling when… and qualitative signaling when…

Mention of NRAS G12D and KRAS G12D does not mention the study by Haigis et al., Nat Genetics, 2008. Is that study not relevant?

When discussing Figure 3A and Figure 3 —figure supplement 2-5, the authors describe a progressive change. Do they simply mean more genes that are significantly different? Or do the identities of genes progressively change? IT is difficult to dissect finer details on the genes from the plot.

In general, increased clarity would be helpful, and more objective descriptions of findings without stronger conclusions than are justified would be helpful.

The main experiment this reviewer would like to see is measurements of RAS-GTP in the MEFs, and – if possible (very likely not) – in specific tissues. (Are mouse macrophages or immune cells not isolatable for measurement?) Would western blots of RAF, MEK, or ERK phosphorylation not be useful to assess signaling? Qualitative differences are of course possible between G12D and Q61R, but it would seem that one could at least evaluate if natural and common codons alter expression of Kras enough to result in changes to signaling downstream from the same mutant (G12D or Q61R).

*Reviewer #3 (Recommendations for the authors):*

The manuscript by Erdogan et al., asks a straightforward question and tightly focusses on answering that question. Further, this manuscript benefits from being very well written and clearly stating the limitations of this study. The authors ask whether the mutational prevalence of KRAS mutations observed in human tumors is due to qualitative (which pathways are engaged) or quantitative (the amplitude of signal) signaling. The authors chose to use a variation on a mouse model that the Counter Lab has previously utilized to approach this question. By using genetically engineered mouse models that have had the first three coding exons of the KRAS locus exchanged to use only rare or common codons to regulate KRAS protein expression levels, and by using two distinct mutations, the common G12D and the rare Q61R, under control of the Rosa26 promoter, this system allowed for mutant protein expression in a broad range of tissues. By measuring the overall GTP-bound levels, a measure of RAS activity, the authors achieved their goal of driving a step-wise increase of total GTP-bound KRAS in the four different models. They reason that quantitative signaling can be observed by changing the expression levels of the same mutant while quantitative signaling is observed by expression of two different mutants that have similar activity levels (KrasnatQ61R vs KrascomG12D).

One limitation to this study is the use of mouse models where the Kras locus has been heavily modified. Further, the use of rare and common codons could have tissue specific effects. However, the authors reason that each mouse has been subjected to the same types of manipulations and RAS function was verified in lung tissue. Because of these limitations, the authors do caution that the degree of signaling cannot be readily compared to previous studies but that these mice can be compared to each other.

It is accepted that the KRAS Q61R mutation has greater overall GTP activity compared to G12D. This was shown in this manuscript using GTP pulldown assays and an ELISA based luciferase assay, showing increased GTP loading in the Q61R mutant relative to G12D and increased expression resulted in increased GTP-bound KRAS. A CRAF RBD pulldown confirmed increased GTP loading. While RBD-pulldowns can be affected by the binding affinity of the RAS mutants to the RBD domain, the use of two distinct assays helps alleviate that concern.

After confirming the model system was behaving as expected, the authors looked at lung tumors and found that increasing activity directly correlated with increasing tumorigenesis, in support of quantitative signaling playing a significant role in tumor formation. However, they also observed that the type of lesions that developed were dependent on the specific KRAS mutation, indicating that qualitative signaling also dictates tumorigenic potential.

The authors performed bulk RNAseq on the total lung tissue to determine how each KRAS allele altered gene transcription levels. Here, the authors found that a number of transcripts involved in numerous connected pathways, analyzed by GSEA analysis, were modulated by the RAS mutants. Importantly, the total number of upregulated genes correlated with total RAS activity levels, indicating that quantitative signaling drives different transcription programs. It is surprising that there is not more overlap between the different mutant RAS constructs in this system. It would be reasonable to expect active RAS to have a core set of effectors that are independent of quantitative and qualitative signaling. However, the extent to which there is any overlap between RAS mutants or by degree of expression is not broadly discussed.

While the active RAS GTP levels appeared to be dependent on both the KRAS mutation and expression levels, the overall survival of the mice expressing the natural codons of both KRASG12D and Q61R were similar. While survival is a global readout of total oncogenic activity, this result does imply that a certain degree of quantitative signaling is necessary to overcome differences in qualitative signaling, a conclusion that the author's also state. Further, looking at tumorigenesis in 8 different tissue types, the authors observed that in some tissue types, quantitative signaling appears to drive the overall malignancy (hematopoietic), while in others (forestomach), qualitative signaling is a larger driver of malignancy.

In conclusion, this study supports the possibility that qualitative signaling preferences of KRAS mutants may drive tumor initiation, in agreement with several recent studies showing mutation-specific signaling, such as KRAS A146T and KRAS G12R. While this study is limited to only two KRAS mutations, the authors suggest that these differences could drive differential inhibitor sensitivities. However, one hallmark of many RAS mutant cancers is the development of increased RAS activity during tumor development, which implies that quantitative signaling may require greater consideration for therapeutic intervention of established tumors. Further, this result leads one to question how successful mutation-specific therapeutics may be in the clinic in qualitative signaling can be masked by quantitative signaling. However, these results also shed light on tumor initiation, which may be useful in determining early detection markers that may be mutation dependent.

Overall, the manuscript has a central theme and the authors do not deviate from this theme. This results in a highly focused paper that clearly draws its main points while being very aware of the limitations of the approach. With these limitations in mind, this manuscript clearly demonstrates mutation specific signaling in an in vivo model, demonstrating yet again that all KRAS mutations are not created equal.

1. Overall, I note very few problems with this text. However, Figure 1 C needs additional clarification. The methods state that the abcam RAS GTPase ELISA kit was used. The product manual states that this kit detects activated H and KRAS human and HRAS rodent samples, if employed per the manufacturer's kit. Was a different primary used? According to the methods, the FLAG-tagged constructs were Kras, which implies a murine system. Given that the empty vector increases with increasing protein extract, it is likely that this system is detecting the endogenous human GTPases and not the ectopically expressed murine construct. While the supplemental data do agree with the expected results based on the published literature, this figure could possibly be improved.

2. In figure 1, supplement 4 panel B, the KrasnatG12D/+ bar does not match the legend.

3. In Figure 5, the gene name for the PI3Ks are incorrect. The correct gene names for p110 isoforms are PIK3CA, PIK3CD and PIK3CG.

4. While not necessary but could improve Figure 5, could testing which inhibitor class on overall survival in the common codon mutants add improve survival? If done in the Q61 mutant, it could be quick to validate the specificity between the classes of inhibitors.

5. Is it possible that expression levels may alter mutant-specific (qualitative) signaling as well? Would this appear as quantitative? More specifically, what was the degree of overlap within the RAS mutants when changing from natural to rare codons? Were all of the hits observed in the natural codon mice detected in the common codon mice? This might be a more useful figure than the STRING analysis.

6. On page 4, it is mentioned that Q61R has slower GTP exchange and hydrolysis rates than G12D. It is worth pointing out that the Burd et al., reference is NRAS while the Rabara et al., reference compared GAP-mediated hydrolysis. However, G12D is interesting because its GTP exchange rate has been reported to be elevated (PMID 8955068 and PMID 24224811), being faster than G13D in biochemical studies, yet this is never discussed. Thus, the author is not wrong in their statement, but additional references and clarification could be useful.

7. On page 6, the authors wrote “special control of gene expression” although I believe they meant “special control”.

8. On page 9, first full paragraph. Discussing RNAseq data as a direct proxy for protein signaling is not entirely valid. RNA levels, which do indicate transcript levels, do not necessarily mean that signaling in those pathways is elevated.

9. On page 11, first paragraph. “Of note, median survival was significantly decreased in mice with either of the Krascom alleles or with either of the Q61R mutants.” This sentence is not entirely clear. Median survival was decreased compared to what? And didn’t the Q61Rnat and G12Dnat have similar survival rates? Please clarify.

---

## [Author Response]

Essential revisions:1) Based on their discussion, the reviewers are in agreement that use of some method is required to demonstrate that the RNA changes documented in the study result in real changes in protein-level expression of key signaling molecules. The reviewers indicate this could be addressed using various approaches, including 1) total proteome analysis; 2) immunoblots for proteins that are important to the analysis; or 3) RPPA. However, some method of evaluating protein changes must be applied, and the results integrated into the analysis.

As requested, we validate the major finds that the four LSL-Kras alleles generate different levels of active Kras (as assessed by RBD affinity capture) that imparts distinct cellular responses at the protein level (as assessed by RPPA) at the endogenous level and in MEF cultures in vitro and/or actual lung tissue a mere days after oncogene activation in vivo, as summarized below:

1. Confirmed a stepwise increase in activation of the Kras oncoprotein in the allelic LSL-Kras set at the endogenous level by Ras Binding Domain (RBD) affinity capture in MEF cultures in vitro. We generated immortalized mouse embryonic fibroblast (MEF) from two independent embryos (i.e. two biological replicates) from each of the genotypes LSL-Kras^natG12D^/+, LSL-Kras^natQ61R^/+, LSL-Kras^comG12D^/+, and LSL-Kras^comQ61R^/+. All eight cell lines were then stably infected with a Cre-expressing retrovirus and successful recombination confirmed by genomic PCR. Two independent immortalized MEF cell lines derived from LSL-Kras^natG12D^/+ mice but not expressing Cre recombinase were cultured in parallel as a control. Duplicate cultures of each of the ten MEF cell lines were serum-starved overnight and the next day one plate was collected as the serum-starved condition to assay basal Kras activity and another plate was treated with serum for 5 minutes before lysates were collected to assess the effect of serum stimulation. Protein lysates were prepared from these 20 samples, protein concentrations were determined, and equal amount of total protein was subjected to affinity capture with a Ras Binding Domain (RBD) polypeptide. Total and RBD pull-down protein samples were resolved by SDS-PAGE and immunoblotted with an anti-KRAS antibody. This analysis confirmed our previous results from HEK-HT cells (now Figure 1—figure supplement 3 and Figure 1—source data 1), namely that changing rare codons to common increases the total amount of Kras protein detected, and that the Q61R mutant leads to higher levels of Kras-GTP than the G12D mutant. This resulted in a stepwise increase in endogenous Kras-GTP levels, beginning with the control cells lacking detectable GTP-Kras, followed in order of increasing activity, the Cre-activated LSL-Kras^natG12D^/+, LSL-Kras^natQ61R^/+, LSL-Kras^comG12D^/+, and finally LSL-Kras^comQ61R^/+ MEF cultures, which was observed in both the serum-starved and serum-stimulated conditions. These new data are provided in the new panel Figure 1C and Figure 1—source data 3. To ensure reproducibility and rigor, we repeated this experiment (i.e. two technical replicates). These new data are provided as the new Figure 1—figure supplement 4 and Figure 1—source data 3.

2. Confirmed a stepwise increase in activation of the Kras oncoprotein in the allelic LSL-Kras set at the endogenous level by Ras Binding Domain (RBD) affinity capture in lung tissue in vivo. We also performed the above analysis from lung tissue isolated from the same allelic set. Namely, we crossed the LSL-Kras^natG12D^/+, LSL-Kras^natQ61R^/+, LSL-Kras^comG12D^/+, and LSL-Kras^comQ61R^/+ genotypes into a Rosa26-CreERT2 background. Two adult mice from each of the four derived cohorts, as well as the control strain (Rosa26-CreERT2/+), were injected with tamoxifen and seven days later humanely euthanized, their lungs removed, and duplicate samples of derived protein lysates were subjected to RBD affinity capture followed by immunoblot (Figure 1—figure supplement 5A). As in the case of MEF cultures described above, we again observed a stepwise increase in Kras activity consistent with the codon usage and oncogenic mutation in vivo. These new data are provided in the new Figure 1—figure supplement 5B,C and Figure 1—source data 4.

3. Confirmed increased activation of each protein in the MAPK pathway in the two most phenotypically different alleles, LSL-Kras^natG12^ versus LSL-Kras^comQ61R^, at the endogenous level by RPPA in lung tissue in vivo. Cohorts of three Rosa26::CreERT2/+;LSL-Kras^natG12D^/+ versus three Rosa26::CreERT2/+;LSL-Kras^comQ61R^/+ (and as a control, Rosa26::CreERT2/+;Kras+/+) mice comprised of an equal distribution of males and females were injected with tamoxifen at 6 to 8 weeks of age to induce recombination and activation of these inducible Kras alleles. We chose these two genotypes as they exhibit the greatest difference in transcriptional profiles (see Figure 3A,B) and tumorigenesis (see Figures 2,4) and encompass both differences in codon usage and mutation type. Seven days later, all mice were humanely euthanized and the lungs removed at the time of necropsy. We chose the lung both because it was the same tissue for which we previously performed RNAseq analysis (see Figure 3) and because this organ exhibits evidence of both the individual mutations as well as codon bias influencing tumor initiation (see Figure 4F). We chose seven days post-injection for this analysis, both because it is earliest point that we validate robust recombination of the LSL-Kras alleles in order to capture the response of normal cells to the oncoprotein (see Figure 4—figure supplement 2) and, again, because this is the same timepoint we previously performed RNAseq analysis (see Figure 3). Protein lysates were then isolated from lung sections and subjected to RPPA analysis. To ensure reproducibility and rigor, lungs from all mice (three biological replicates) were tested in triplicate (three technical replicates). We note that originally we had four mice in the Rosa26::CreERT2/+;LSL-Kras^natG12D^/+ cohort, but one sample was censored as it had a completely different RPPA profile compared to the other three mice in this cohort. RPPA was performed on 140 proteins/phosphoproteins representative of pathways altered the most in the two extreme transcriptional profiles, namely derived from the lungs of LSL-Kras^G12D^(nat) versus LSL-Kras^Q61R^(com) mice shortly after recombination was induced. Focusing on the MAPK pathway, we report increased phosphorylation of the proteins of this pathway in the Rosa26::CreERT2/+;LSL-Kras^comQ61R^/+ murine lung tissue. Again, in complete concordance with the immunoblot analysis of Kras-GTP levels in MEFs and/or lung tissue, as well as the transcriptome analysis of lung tissue. These new data are provided in the new Figure 1—figure supplement 1 and Supplementary file 1.

4. Confirmed that the transcriptomes manifest in differences at the protein level in the two most different alleles, LSL-Kras^natG12^ versus LSL-Kras^comQ61R^, at the endogenous level by RPPA in lung tissue in vivo. As noted, our RPPA analysis focused on proteins representative of pathways altered in the two extreme transcriptional profiles. Specifically, proteins within the RAS/MAPK, PI3K/AKT/mTOR, Growth, IL/Jak/Stat, DNA repair, Apoptosis, Autophagy, and Senescence pathways (see Figure 3B,C and Figure 3—figure supplement 1C). We report concordance between the transcriptional and proteomic profiles, namely an upregulation in proteins in the MAPK pathway, those involved in stress response and DNA damage, and cytokine activation. These new data are provided in the new Figure 3—figure supplement 9 and Supplementary file 1.

2) In addition, each of the three reviewers made some suggestions regarding phrasing and interpretation. The authors are strongly encouraged to address these.

As requested, all other concerns were positively addressed, either by performing the suggested experiments or significantly revising the text accordingly.

Reviewer #1 (Recommendations for the authors):The authors try to establish biological differences between commonly encountered KRAS mutants. Distinct mutations exhibit tissue tropism (e.g., KRAS G12C in lung, G12D and G12R in pancreas, etc), but the reasons for these distributions have never been fully clarified. By establishing mouse models in which the type of KRAS mutant and its level of expression can be readily controlled, the authors hope to establish whether the type of amount of KRAS mutant expressed will result in distinct outcomes.Major strengths include the extensive body of work from the Counter lab regarding the key role of codon usage in KRAS oncogenicity, the generation of several sets of matched mouse models (two of which are studied in detail, but the others are also of value), and the examination of KRAS effects in many tissues.Weaknesses include the fact (acknowledged by the authors) that the system is highly artificial, in that KRAS is being induced globally at one point in time, that its expression levels are artificially boosted by codon changes that would not occur in vivo (though perhaps might happen if the KRAS gene were amplified, as sometimes happens), and a lack of any protein level data to corroborate the RNA profiling performed in the study.

We thank the reviewer for both their encouraging review and thoughtful suggestions. We note here that while admittedly artificial, whole-body activation of each of the four novel *LSL-Kras* alleles was key to identify the sensitivities of different tissues to each mutant. We also note here that changing rare codons to common is essentially a hypermorphic allele, which is a staple of mouse genetics. Nevertheless, we see your point, so if there are specific recommendations to the text beyond what we have already included to better describe the limitations of these two genetic approaches, we would be glad to include them- we want to get this right! As requested, please see comment 1 below describing RPPA analysis.

It would strengthen the work if the authors would back up some of the key RNA signatures with western blots of RPPA-type data.

As requested, cohorts of three Rosa26::CreERT2/+;LSL-Kras^natG12D^/+ versus three Rosa26::CreERT2/+;LSL-Kras^comQ61R^/+ (and as a control, Rosa26::CreERT2/+;Kras+/+) mice comprised of an equal distribution of males and females were injected with tamoxifen at 6 to 8 weeks of age to induce recombination and activation of these inducible Kras alleles. We chose these two genotypes as they exhibit the greatest difference in transcriptional profiles (see Figure 3A,B) and tumorigenesis (see Figures 2,4). Seven days later, all mice were humanely euthanized and the lungs removed at the time of necropsy. We chose the lung both because it was the same tissue for which we previously performed RNAseq analysis (see Figure 3) and because this organ exhibits evidence of both the individual mutations as well as codon bias influencing tumor initiation (see Figure 4F). We chose seven days post-injection for this analysis, both because it is earliest point that we validate robust recombination of the LSL-Kras alleles in order to capture the response of normal cells to the oncoprotein (see Figure 4—figure supplement 2) and, again, because this is the same timepoint we previously performed RNAseq analysis (see Figure 3). Protein lysates were then isolated from lung sections and subjected to RPPA analysis. To ensure reproducibility and rigor, lungs from all mice (three biological replicates) were tested in triplicate (three technical replicates). We note that originally we had four mice in the Rosa26::CreERT2/+;LSL-Kras^natG12D^/+ cohort, but one sample was censored as it had a completely different RPPA profile compared to the other three mice in this cohort. RPPA was performed on 140 proteins/phosphoproteins representative of pathways altered the most in the two extreme transcriptional profiles, namely derived from the lungs of LSL-Kras^G12D^(nat) versus LSL-Kras^Q61R^(com) mice shortly after recombination was induced. Focusing on the MAPK pathway, we report increased phosphorylation of the proteins of this pathway in the Rosa26::CreERT2/+;LSL-Kras^comQ61R^/+ compared to the Rosa26::CreERT2/+;LSL-Kras^natG12D^/+ murine lung tissue. These new data are provided in the new Figure 1—figure supplement 7 and Supplementary file 1. Additionally, we also included analysis of proteins/phosphoproteins from pathways that transcriptome analysis identified as being upregulated in the lung upon activation of the Kras^comQ61^ compared to the Kras^natG12D^ allele. Bulk transcriptome analysis revealed that activation of the Kras^comQ61R^ allele had all features of a potent oncogenic signaling leading to hyperproliferation and oncogenic-induced stress. Namely, GSEA hallmarks indicative of high oncogenic signaling (KRAS Signaling UP) and unrestrained proliferation (MYC Targets V1, E2F Targets), leading to reactive oxygen species (Oxidative Phosphorylation, Reactive Oxygen Species Pathway) and a DNA damage response (DNA Repair, P53 Pathway, G2M Checkpoint) followed by apoptosis (Apoptosis) and inflammation (Inflammatory Response, IL6 JAK STAT3 Signaling, TNFa Signaling via NF-kb, Allograft Rejection) (see Figure 3). Comparing the level of proteins/phosphoprotein by RPPA of Kras^natG12D^ to Kras^comQ61^ (normalized to the control strain) revealed a clear increase in the latter in RAS/MAPK (as already noted), PI3K/AKT, Growth, DNA repair, senescence/autophagy/apoptosis, and IL/Jak/Stat, thereby validating the transcriptome analysis. These new data are included as Figure 3—figure supplement 9 and Supplementary file 1.

It could be instructive if the authors performed exon sequencing on some of their tumors. They saw no pancreatic tumors, which is a bit surprising, though it was perhaps because tp53 was not also mutated. One wonder what mutations might have occurred in those tissues where tumors were seen, such as lung.

This is a really great suggestion, and one that we will most certainly perform in the future as it would be ever-so-interesting to see if the type of foundational Kras mutation influences the nature of mutations selected to drive tumor progression. However, since the current study focuses on tumor initiation and exploring this new idea will be a major undertaking to do correctly, we propose to address this new concept in the future. Thank you for this idea! With regards to why pancreatic tumors fail to develop in the genetic backgrounds we employed in this study, we are not the first to observe this. Indeed, we now clarify in the revised text that Parikh, *et al.,* (PMID: 22532587) and Van der Weyden, *et al.,* (PMID:21381032) both published that inducing recombination of an *LSL-Kras^G12D^* allele in the identical *Rosa26-CreERT2* background in adult mice did *not* generate pancreatic tumors. Instead, we note here that an oncogenic *Kras* allele induces pancreatic tumors if activated either during development or in adult when accompanied by either pancreatitis or loss of tumor suppressors (Guerra and Barbacid, Mol Oncol, 2013, PMID: 23506980), but not in adults without further perturbations.

Reviewer #2 (Recommendations for the authors):Within this manuscript, the authors utilize mouse genetic approaches to modulate both which Kras mutation is expressed and how much of that Kras mutant is expressed. By utilizing different codons, they can have the same Kras mutation expressed at either a low level (due to rare codon utilization) or at a higher level (through common codon utilization). This is a clever strategy that this group pioneered for studying RAS signaling, and by now comparing two different Kras mutants (G12D and Q61R) they can also introduce additional quantitative and qualitative variation.The authors define quantitative variation as variation in the intensity of active Kras-GTP, and qualitative variation as how the Kras-GTP may activate different downstream effector proteins. This is a well-defined concept that helps provide a structure to their analysis.The genetic approach, and the setting of definitions, are major strengths as the authors dissect this complicated problem in a logical, systematic manner. Additional strengths include studying how the different Kras strains differ for different tissue phenotypes.However, there are a few weaknesses to the manuscript in its current form. Although the authors' hypotheses about quantitative and qualitative variation are likely true, at times the authors appear to interpret their data more strongly than the data seem to permit. In many of these cases, a rephrasing that describes their findings objectively would be valuable because the data – at minimum – highlight the need for these quantitative and qualitative factors to be better considered within the field.One seemingly striking weakness is that the allelic series does not seem to be sufficiently well demonstrated to have the progressive increase in mutant strengths. Another, and partially overlapping weakness, is that the authors seem to invoke tissue-specific differences when convenient and to ignore them when convenient.The field of RAS biology has made great progress ignoring the potential effects of signal intensity and mutant-to-mutant variation, but the reinvigoration of the field in recent years has revealed important mutant-to-mutant specific differences and instances where intensity matters. A deeper understanding of these factors, and improved tools to study these differences, are both important to the field. The authors are here advancing tools and approaches that could be used in future studies and are providing a foundation for future studies. Thus, the work is likely to be impactful and influential.

We thank the reviewer for both their encouraging review and thoughtful suggestions. As requested, we extensively revised the text to better describe the data and our interpretations thereof, especially with regards to different interpretations of qualitative signaling, although if there are specific sentences or points the reviewer would like us to change or add, we would be glad to abide by their suggestions. As requested, we also positively addressed the issue of allelic strength in reply to your comment 1 below.

One concern is the demonstration of the progressive quantitative strength of the allelic series. Even though a mouse genetic approach is utilized for the manuscript, the comparisons of Kras mutant strength is made with transiently transfected HEK-HT cells. Could the authors not do this in the MEFs they utilize, which would be endogenous expression from these cells derived from the genetically engineered mice?

As requested, we generated immortalized mouse embryonic fibroblast (MEF) from two independent embryos (i.e. two biological replicates) from each of the genotypes LSL-Kras^natG12D^/+, LSL-Kras^natQ61R^/+, LSL-Kras^comG12D^/+, and LSL-Kras^comQ61R^/+. All eight cell lines were then stably infected with a Cre-expressing retrovirus and successful recombination confirmed by genomic PCR. Two independent immortalized MEF cell lines derived from LSL-Kras^natG12D^/+ mice but not expressing Cre recombinase were cultured in parallel as a control. Duplicate cultures of each of the ten MEF cell lines were serum-starved overnight and the next day one plate was collected as the serum-starved condition to assay basal Kras activity and another plate was treated with serum for 5 minutes before lysates were collected to assess the effect of serum stimulation. Protein lysates were prepared from these 20 samples, protein concentrations were determined, and equal amount of total protein was subjected to affinity capture with a Ras Binding Domain (RBD) polypeptide. Total and RBD pull-down protein samples were resolved by SDS-PAGE and immunoblotted with an anti-KRAS antibody. This analysis confirmed our previous results from HEK-HT cells (now Figure 1—figure supplement 3 and Figure 1—source data 1), namely that changing rare codons to common increases the total amount of Kras protein detected, and that the Q61R mutant leads to higher levels of Kras-GTP than the G12D mutant. This resulted in a stepwise increase in endogenous Kras-GTP levels, beginning with the control cells lacking detectable GTP-Kras, followed in order of increasing activity, the Cre-activated LSL-Kras^natG12D^/+, LSL-Kras^natQ61R^/+, LSL-Kras^comG12D^/+, and finally LSL-Kras^comQ61R^/+ MEF cultures, which was observed in either the serum-starved or serum-stimulated conditions. These new data are provided in the new panel Figure 1C and Figure 1—source data 3. To ensure reproducibility and rigor, we repeated this experiment (i.e. two technical replicates). These new data are provided as the new Figure 1—figure supplement 4 and Figure 1—source data 3. Additionally, please see our response to your comment 10b, as we also repeated this analysis on the lungs of mice from these four backgrounds, the results of which mirrored those from MEF cultures.

The authors later claim, and acknowledge, tissue differences in factors that can contribute to the overall activation of Kras, and the authors also acknowledge qualitative differences. It seems reasonable to consider that there could be tissue-specific differences in which GEFs are activated, and that active GEFs might act upon the G12D and Q61R mutants differently – just as effector proteins may bind the mutants differently. The authors seem to be assuming the relative strengths observed are invariant between cell/tissue types and that their observations are either reflective of these quantitative differences, or that if they differ, that they must be qualitative. But why can't there be qualitative upstream differences that impact the quantity of Kras-GTP? In other words – quantitative and qualitative influences still seem to be intertwined in this system. The authors may want to state that these are strong (and unproven) assumptions, and that their interpretations are based upon these assumptions.

As requested, we revised the text to highlight this point.

The authors refer to "normal cells" in the introduction; they appear to be referring to HEK-HT cells. It seems misleading to refer to them as "normal". The authors should be more precise.

We revised the text so that “normal cells” does not refer to HEK-HT cells.

The title seems to be too strongly stated. Mutation patterns also arise from the mutagens (cigarette smoking in lung causing G12C, etc.). Additionally, the authors do not show this or prove this – this is a speculation down the road from their findings, by my interpretation of their study. A more objective title would be more appropriate.

As requested, we changed the title to “Genetically manipulating endogenous Kras levels and oncogenic mutations in vivo influences tissue patterning of murine tumorigenesis”.

The authors state "determining the immediate response of normal cells to different Kras mutations in vivo thus holds the key to understanding the mutational patterning of this oncogene". The logic to make such a strong conclusion is not apparent. This rather seems to be a hypothesis, that studying the immediate response may offer some insight into mutational patterning. Several specific details in the sentence appear overly strong. Can this be rephrased?

As requested, we rephrased this sentence.

The sentence "In this way, tissues sensitive to quantitative signaling…". This is important, as it highlights the authors interpretation of their data (and possible flaws with this assumption are mentioned above). Could the authors rephrase this a bit more mildly to describe that the authors interpret their data to indicate quantitative signaling when… and qualitative signaling when…

As requested, we rephrased this and other similar sentences to address this point.

Mention of NRAS G12D and KRAS G12D does not mention the study by Haigis et al., Nat Genetics, 2008. Is that study not relevant?

As requested, while the focus of this study was to compare the same isoform with two different mutations and expression levels to parse out the potential contribution of quantitative versus qualitative signaling (with the caveats of this approach noted), we agree that there is great value in expanding our interpretations to isoforms, and now cite this informative study in the introduction.

When discussing Figure 3A and Figure 3 —figure supplement 2-5, the authors describe a progressive change. Do they simply mean more genes that are significantly different? Or do the identities of genes progressively change? IT is difficult to dissect finer details on the genes from the plot.

As requested, we rephrased the explanation emphasizing that the total number of different genes differentially expressed in the mutant alleles in comparison to the WT allele gradually increase as the RAS activity (as defined by RBD pull-downs) of the oncoprotein encoded by these alleles increase (i.e. the identity of the genes changes). As requested, to make this easier to see we include full-page volcano plots in Figure 3—figure supplements 2-5. We also clarify that genes differentially expressed in lung tissue with the lowest level of Kras activity remain activated in when Kras is more highly activated (i.e. expression of the same genes throughout the genotypes). As requested, to clarify this point we also add a Venn diagram breaking down the number of genes shared between all possible comparisons in the new Figure 3—figure supplement 6.

In general, increased clarity would be helpful, and more objective descriptions of findings without stronger conclusions than are justified would be helpful.

As requested, we extensively revised the text accordingly.

The main experiment this reviewer would like to see is measurements of RAS-GTP in the MEFs, and – if possible (very likely not) – in specific tissues. (Are mouse macrophages or immune cells not isolatable for measurement?)

As requested, we performed the suggested experiment in MEF cultures which a stepwise increase in endogenous Kras-GTP levels, beginning with the control cells lacking detectable GTP-Kras, followed in order of increasing activity, the Cre-activated LSL-Kras^natG12D^/+, LSL-Kras^natQ61R^/+, LSL-Kras^comG12D^/+, and finally LSL-Kras^comQ61R^/+ MEF cultures. Please see our reply to your comment 1 for details and Figure 1C, Figure 1—figure supplement 4, and Figure 1—source data 3. As requested, we additionally performed RBD affinity capture on lysates derived from lung tissue isolated from the same allelic set. Namely, we crossed the LSL-Kras^natG12D^/+, LSL-Kras^natQ61R^/+, LSL-Kras^comG12D^/+, and LSL-Kras^comQ61R^/+ genotypes into a Rosa26-CreERT2 background. Two adult mice from each of the four derived cohorts, as well as the control strain (Rosa26-CreERT2/+), were injected with tamoxifen and seven days later humanely euthanized, their lungs removed, and duplicate samples of derived protein lysates were subjected to RBD affinity capture followed by immunoblot (Figure 1—figure supplement 5A). As in the case of MEF cultures, we again observed a stepwise increase in Kras activity consistent with the codon usage and oncogenic mutation in vivo. These new data are provided in the new Figure 1—figure supplement 5 and Figure 1—source data 4.

Would western blots of RAF, MEK, or ERK phosphorylation not be useful to assess signaling? Qualitative differences are of course possible between G12D and Q61R, but it would seem that one could at least evaluate if natural and common codons alter expression of Kras enough to result in changes to signaling downstream from the same mutant (G12D or Q61R).

As requested, cohorts of three Rosa26::CreERT2/+;LSL-Kras^natG12D^/+ versus three Rosa26::CreERT2/+;LSL-Kras^comQ61R^/+ (and as a control, Rosa26::CreERT2/+;Kras+/+) mice comprised of an equal distribution of males and females were injected with tamoxifen at 6 to 8 weeks of age to induce recombination and activation of these inducible Kras alleles. We chose these two genotypes as they exhibit the greatest difference in transcriptional profiles (see Figure 3A,B) and tumorigenesis (see Figures 2,4) as well as encompass differences in both codon usage and mutation type. Seven days later, all mice were humanely euthanized and the lungs removed at the time of necropsy. We chose the lung both because it was the same tissue for which we previously performed RNAseq analysis (see Figure 3) and because this organ exhibits evidence of both the individual mutations as well as codon bias influencing tumor initiation (see Figure 4F). We chose seven days post-injection for this analysis, both because it is earliest point that we validate robust recombination of the LSL-Kras alleles in order to capture the response of normal cells to the oncoprotein (see Figure 4—figure supplement 2) and, again, because this is the same timepoint we previously performed RNAseq analysis (see Figure 3). Protein lysates were then isolated from lung sections and subjected to RPPA analysis. To ensure reproducibility and rigor, lungs from all mice (three biological replicates) were tested in triplicate (three technical replicates). We note that originally we had four mice in the Rosa26::CreERT2/+;LSL-Kras^natG12D^/+ cohort, but one sample was censored as it had a completely different RPPA profile compared to the other three mice in this cohort. RPPA was performed on 140 proteins/phosphoproteins representative of pathways altered the most in the two extreme transcriptional profiles, namely derived from the lungs of LSL-Kras^G12D^(nat) versus LSL-Kras^Q61R^(com) mice shortly after recombination was induced. Focusing on the MAPK pathway proteins in this analysis, we report increased phosphorylation of the proteins of this pathway in the Rosa26::CreERT2/+;LSL-Kras^comQ61R^/+ compared to the Rosa26::CreERT2/+;LSL-Kras^natG12D^/+ mice lung tissue. Again, in complete concordance with the immunoblot analysis of Kras-GTP levels in MEFs and lung tissue, as well as the transcriptome of these cells. These new data are provided in the new Figure 1—figure supplement 7 and Supplementary file 1.

Reviewer #3 (Recommendations for the authors):The manuscript by Erdogan et al., asks a straightforward question and tightly focusses on answering that question. Further, this manuscript benefits from being very well written and clearly stating the limitations of this study. The authors ask whether the mutational prevalence of KRAS mutations observed in human tumors is due to qualitative (which pathways are engaged) or quantitative (the amplitude of signal) signaling. The authors chose to use a variation on a mouse model that the Counter Lab has previously utilized to approach this question. By using genetically engineered mouse models that have had the first three coding exons of the KRAS locus exchanged to use only rare or common codons to regulate KRAS protein expression levels, and by using two distinct mutations, the common G12D and the rare Q61R, under control of the Rosa26 promoter, this system allowed for mutant protein expression in a broad range of tissues. By measuring the overall GTP-bound levels, a measure of RAS activity, the authors achieved their goal of driving a step-wise increase of total GTP-bound KRAS in the four different models. They reason that quantitative signaling can be observed by changing the expression levels of the same mutant while quantitative signaling is observed by expression of two different mutants that have similar activity levels (KrasnatQ61R vs KrascomG12D).One limitation to this study is the use of mouse models where the Kras locus has been heavily modified. Further, the use of rare and common codons could have tissue specific effects. However, the authors reason that each mouse has been subjected to the same types of manipulations and RAS function was verified in lung tissue. Because of these limitations, the authors do caution that the degree of signaling cannot be readily compared to previous studies but that these mice can be compared to each other.It is accepted that the KRAS Q61R mutation has greater overall GTP activity compared to G12D. This was shown in this manuscript using GTP pulldown assays and an ELISA based luciferase assay, showing increased GTP loading in the Q61R mutant relative to G12D and increased expression resulted in increased GTP-bound KRAS. A CRAF RBD pulldown confirmed increased GTP loading. While RBD-pulldowns can be affected by the binding affinity of the RAS mutants to the RBD domain, the use of two distinct assays helps alleviate that concern.After confirming the model system was behaving as expected, the authors looked at lung tumors and found that increasing activity directly correlated with increasing tumorigenesis, in support of quantitative signaling playing a significant role in tumor formation. However, they also observed that the type of lesions that developed were dependent on the specific KRAS mutation, indicating that qualitative signaling also dictates tumorigenic potential.The authors performed bulk RNAseq on the total lung tissue to determine how each KRAS allele altered gene transcription levels. Here, the authors found that a number of transcripts involved in numerous connected pathways, analyzed by GSEA analysis, were modulated by the RAS mutants. Importantly, the total number of upregulated genes correlated with total RAS activity levels, indicating that quantitative signaling drives different transcription programs. It is surprising that there is not more overlap between the different mutant RAS constructs in this system. It would be reasonable to expect active RAS to have a core set of effectors that are independent of quantitative and qualitative signaling. However, the extent to which there is any overlap between RAS mutants or by degree of expression is not broadly discussed.While the active RAS GTP levels appeared to be dependent on both the KRAS mutation and expression levels, the overall survival of the mice expressing the natural codons of both KRASG12D and Q61R were similar. While survival is a global readout of total oncogenic activity, this result does imply that a certain degree of quantitative signaling is necessary to overcome differences in qualitative signaling, a conclusion that the author's also state. Further, looking at tumorigenesis in 8 different tissue types, the authors observed that in some tissue types, quantitative signaling appears to drive the overall malignancy (hematopoietic), while in others (forestomach), qualitative signaling is a larger driver of malignancy.In conclusion, this study supports the possibility that qualitative signaling preferences of KRAS mutants may drive tumor initiation, in agreement with several recent studies showing mutation-specific signaling, such as KRAS A146T and KRAS G12R. While this study is limited to only two KRAS mutations, the authors suggest that these differences could drive differential inhibitor sensitivities. However, one hallmark of many RAS mutant cancers is the development of increased RAS activity during tumor development, which implies that quantitative signaling may require greater consideration for therapeutic intervention of established tumors. Further, this result leads one to question how successful mutation-specific therapeutics may be in the clinic in qualitative signaling can be masked by quantitative signaling. However, these results also shed light on tumor initiation, which may be useful in determining early detection markers that may be mutation dependent.Overall, the manuscript has a central theme and the authors do not deviate from this theme. This results in a highly focused paper that clearly draws its main points while being very aware of the limitations of the approach. With these limitations in mind, this manuscript clearly demonstrates mutation specific signaling in an in vivo model, demonstrating yet again that all KRAS mutations are not created equal.

We thank the reviewer for both their encouraging review and thoughtful suggestions.

1. Overall, I note very few problems with this text. However, Figure 1 C needs additional clarification. The methods state that the abcam RAS GTPase ELISA kit was used. The product manual states that this kit detects activated H and KRAS human and HRAS rodent samples, if employed per the manufacturer's kit. Was a different primary used? According to the methods, the FLAG-tagged constructs were Kras, which implies a murine system. Given that the empty vector increases with increasing protein extract, it is likely that this system is detecting the endogenous human GTPases and not the ectopically expressed murine construct. While the supplemental data do agree with the expected results based on the published literature, this figure could possibly be improved.

As requested, we clarify in the Materials and methods the specific antibody used (which was not a different antibody than that provided by this kit). As requested, we clarify in text that the ectopic Kras expressed in HEK-HT cells was derived from a FLAG-tagged murine Kras cDNA. As requested, we note here that the manufacturer indicated that this kit was validated with full-length human Kras/Hras and murine Hras, but not murine Kras. Given this, we tested and found that this kit captured FLAG-tagged murine Kras ectopically expressed in HEK-HT cells, while no signal was present in the vector control cells. This finding is consistent with the kit detecting the over-expressed murine Kras protein in the GTP-bound state.

2. In figure 1, supplement 4 panel B, the KrasnatG12D/+ bar does not match the legend.

As requested, we first thank the reviewer for catching this oversight and have corrected the figure legend.

3. In Figure 5, the gene name for the PI3Ks are incorrect. The correct gene names for p110 isoforms are PIK3CA, PIK3CD and PIK3CG.

As requested, we first thank the reviewer for catching this and have corrected the nomenclature in this figure.

4. While not necessary but could improve Figure 5, could testing which inhibitor class on overall survival in the common codon mutants add improve survival? If done in the Q61 mutant, it could be quick to validate the specificity between the classes of inhibitors.

This would be a natural progression of this study. However, since the current study focuses on tumor initiation, we propose to address the concept of targeting specific mutants in mice would be best suited for future exploration.

5. Is it possible that expression levels may alter mutant-specific (qualitative) signaling as well? Would this appear as quantitative?

As requested, we include this possibility in the text.

More specifically, what was the degree of overlap within the RAS mutants when changing from natural to rare codons? Were all of the hits observed in the natural codon mice detected in the common codon mice? This might be a more useful figure than the STRING analysis.

As requested, we compared NatG12D to ComG12D, and found all seven differentially expressed genes in the NatG12D with respect to the WT allele were also differentially expressed in the ComG12D transcriptome with very similar log2FC and adjusted *p-*values. Five of these genes were differentially expressed in all alleles. Comparing NatQ61R to ComQ61R, we found 12 out of 15 differentially expressed genes in the NatQ61R allele were also differentially expressed in the ComQ61R transcriptome. These new data are now provided in the form of a Venn diagram in Figure 3—figure supplement 6.

6. On page 4, it is mentioned that Q61R has slower GTP exchange and hydrolysis rates than G12D. It is worth pointing out that the Burd et al., reference is NRAS while the Rabara et al., reference compared GAP-mediated hydrolysis. However, G12D is interesting because its GTP exchange rate has been reported to be elevated (PMID 8955068 and PMID 24224811), being faster than G13D in biochemical studies, yet this is never discussed. Thus, the author is not wrong in their statement, but additional references and clarification could be useful.

As requested, we indicate that the Burd *et al.,* reference refers to NRAS and that the Rabara *et al.,* reference compared GAP-mediated hydrolysis. Since the focus of this study is between G12D and Q61R mutations we elected not to bring in comparisons with G13 mutations. However, if we have missed the point of the reviewer, we are glad to make further changes to this section.

7. On page 6, the authors wrote "special control of gene expression" although I believe they meant "special control".

As requested, we first thank the reviewer for catching this and change this to spatial.

8. On page 9, first full paragraph. Discussing RNAseq data as a direct proxy for protein signaling is not entirely valid. RNA levels, which do indicate transcript levels, do not necessarily mean that signaling in those pathways is elevated.

As requested, we rephrased the rationale to use RNAseq as a way to capture changes to cells at the level of the transcriptome.

9. On page 11, first paragraph. "Of note, median survival was significantly decreased in mice with either of the Krascom alleles or with either of the Q61R mutants." This sentence is not entirely clear. Median survival was decreased compared to what? And didn't the Q61Rnat and G12Dnat have similar survival rates? Please clarify.

As requested, we revised the text to indicate that the median survival of mice in which either of the *LSL-Kras^com^* allele were activated was lower than when either of the *LSL-Kras^nat^* alleles were activated. As requested, we revised the text to indicate that the median survival of mice in which the *LSL-Kras^comQ61R^* allele was activated was lower than when the *LSL-Kras^comG12D^* allele was activated. Finally, and as requested, we revised the text to indicate that the median survival of mice in which the *LSL-Kras^natQ61R^* allele was activated trended, but did not reach significance as being lower than when the *LSL-Kras^natG12D^* allele was activated.